# PREDICTING EPISODIC STRUCTURE FROM OVERLAPPING INPUT IN BINARY NETWORKS WITH HOMEOSTASIS

## ABSTRACT

How neural networks process overlapping input patterns is a fundamental question in both neuroscience and artificial intelligence. Traditionally, overlaps in neural activity are viewed as interference, requiring separation for better performance. However, an alternative perspective suggests that these overlaps may encode meaningful semantic relationships between concepts. In this paper, we propose a framework where persistent overlap between episodic patterns represent semantic components across episodic experiences, and the statistics of these overlaps how each semantic concept relates to others.

To explore this idea, we introduce an Episode Generation Protocol (EGP) that defines a mapping between the semantic structure of episodes and input pattern generation. Paired with our EGP, we use Homeostatic Binary Networks (HBNs), a simplified yet biologically-inspired model incorporating key features such as adjustable inhibition, Hebbian learning, and homeostatic plasticity.

Our contributions are threefold: (1) We formalize a link between episodic semantics and neural patterns through our EGP. This EGP can be used for systematic study of semantic learning in artificial neural networks. (2) We introduce HBNs as an analytically tractable network that extracts semantic structure in its internal model (3) We show that HBNs align their performance with Maximum A Posteriori and Maximum Likelihood Estimation strategies depending on the homeostatic regime. Similarly, we provide an example of how our EGP can be used as an experimental protocol in neuroscience to make different models of learning compete.

## 1 INTRODUCTION

Understanding how neural networks learn and process overlapping patterns is a central question in neuroscience and artificial intelligence (AI) (O'Reilly, 2000). Overlapping patterns are many times viewed as sources of interference or noise, which need to be separated in order to avoid a degraded performance (French, 1999; Goodfellow et al., 2013). Another view is that overlaps might not only represent interference but also encode meaningful semantic relationships. In particular, overlaps are proposed to encode the similarities between the concepts represented by different patterns (De Falco et al., 2016; Gastaldi et al., 2021; Gastaldi & Gerstner, 2024) (see Fig. 1A). However, this outlook typically assumes that each pattern separately represents a concept, and that the overlap is a consequence of the concepts being semantically related. A different interpretation, which we explore here, is that activity patterns correspond to the full content of episodes, and the overlaps between patterns represent common concepts in the episodes encoded by both patterns (Fig. 1B). Understanding the meaning of these overlaps is tightly related to the notion of semantics. In this sense, the interplay between semantics and learning has puzzled researchers in AI and cognitive sciences for decades, with a recovered popularity in the relatively recent years (Saxe et al., 2019; Chrysanthidis et al., 2022; Ben-Shaul et al., 2023; Ravichandran et al., 2024).

In order to study how semantics impact learning, we propose an Episode Generation Protocol (EGP), which samples input with a well-defined semantic structure. Paired with this EGP, we introduce *Homeostatic Binary Networks* (HBNs), a minimal model that incorporates biologically-inspired elements such as adjustable inhibition, Hebbian learning, and homeostatic plasticity. Due

to their simplicity, for certain regimes, one can obtain the closed-form trajectories of semantically-labeled populations of weights in these networks.

The main contributions of this work are:

- Proposing an Episode Generation Protocol (EGP) with an associated Semantic Structure, which allows testing semantic learning in artificial neural networks.
- Presenting Homeostatic Binary Networks (HBNs). This simplification of biologically-inspired neural networks allows a direct link between a formally defined Semantic Structure and its learning dynamics.
- Using the EGP to obtain different behavioural signatures of a plethora of models used in neuroscience, relating them to responses in prototype learning and masked input prediction.

## 2 RESULTS

### 2.1 EPISODE GENERATION PROTOCOL (EGP) FOR SEMANTIC LEARNING

To test neural networks' ability to extract semantic relationships from inputs, we propose an *Episode Generation Protocol (EGP)*. This framework connects episodic patterns to semantic structure, enabling systematic testing of learning systems. The EGP defines episodes as combinations of *concepts* (e.g., *Italy* and *pizza*) across *attributes* (e.g., *place* and *food*) (Fig. 1C, left), with sampling probabilities specifying the stochastic relationships between concepts (Fig. 1C, center).

Each episode maps to an input pattern where concepts activate subsets of neurons, forming a list of *K One-Hot* encodings for each attribute. Having multiple neurons coding for each concept allows further noise injection without totally erasing the original meaning. Here, variability is introduced by randomly choosing $N_{\text{swap}}/2$ active neurons, and $N_{\text{swap}}/2$ inactive neurons, and flipping their activity. This results in a total of $N_{\text{swap}}$ neurons stochastically changing their original state, but overall preserving the total number of active neurons. The process $episode \longrightarrow input\ pattern \longrightarrow noise$ defines the *episode-input mapping* of the EGP (Fig. 1C, right), which could however take alternative forms to the one proposed here.

The *semantic structure* of the EGP is formalized as a matrix $SS$, where each entry $SS_{ij}$ represents the conditional probability of episode concept $j$ given episode concept $i$ (Fig. 1E). For example, in an EGP where *Italy* only appears with *pizza*, the probability of *pizza* given *Italy* is 1, while *Italy* given *pizza* is 0.5 due to the shared occurrence of *pizza* with *France* (Fig. 1D). This structure encodes asymmetries and causal relationships between concepts, extending beyond standard methods such as Representational Similarity Analysis (Kriegeskorte et al., 2008; Schapiro et al., 2017a) or Point-wise Mutual Information (Fano, 1968; De Falco et al., 2016).

Importantly, the EGP generalizes (and formalizes the semantic structure of) previously used overlapping input protocols, providing a general framework for testing semantic learning. In Appendix A.3, we re-frame prior datasets (Schapiro et al., 2017b; Singh et al., 2022; Fung & Fukai, 2023) within this protocol, highlighting its applicability to formalize existing experiments.

### 2.2 SEMANTIC LEARNING IN RECURRENT NEURAL NETWORKS

We start examining how the semantic structure of an EGP influences the learned connectivity in various recurrent network architectures. Recurrent neural networks (RNNs) are networks in which every unit receives input from the rest, and are widely used in machine learning and neuroscience. Here, we train three different types of RNNs: a Hopfield-Tsodyks network (Tsodyks & Feigel'man, 1988), a Boltzmann Machine (Ackley et al., 1985), and a feed-forward network (Appendix B) where the output has the same size as the input. We show the connectivity both *early* and *late* in training of the feed-forward network.

The networks are trained on episodes sampled from an EGP with a simple yet asymmetric semantic structure (Fig. 2A-B). After training, synaptic strength is normalized between 0 and 1, (Fig. 2C). For each network, we calculate the correlation coefficient $r_{\text{sem}}$ between this normalized connectivity matrix and the semantic structure of the EGP (maximum value of cosine similarity considering both the original weight matrix and its transpose).

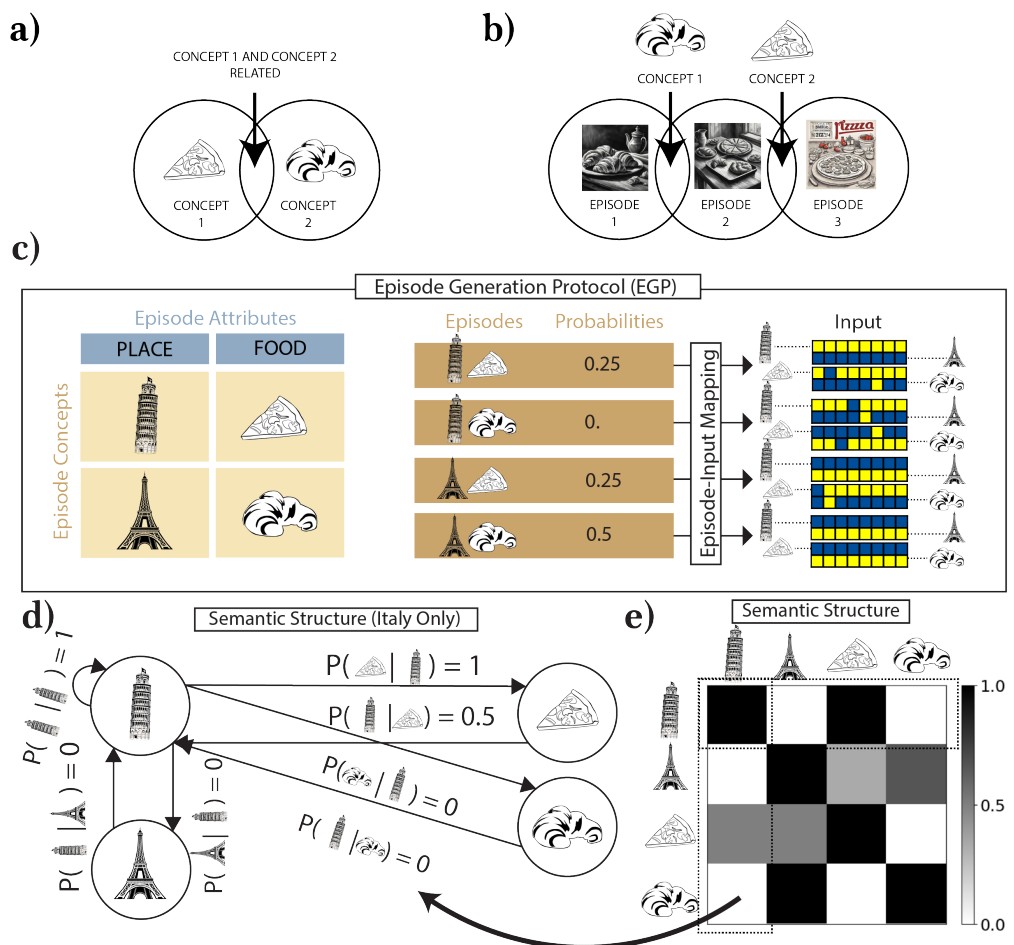

Figure 1: **A:** Overlapping neuronal activity patterns as proposed in De Falco et al. (2016); Gastaldi et al. (2021); Gastaldi & Gerstner (2024). Here each pattern corresponds to a concept, and the overlap between patterns is a result of the patterns being semantically related. **B:** Our proposed view of engram overlaps. Here, activity patterns code for episodes, and the persistent overlap between patterns are the concepts. **C:** Schematics of our Episode Generation Protocol (EGP). An EGP contains (**left**) episode attributes (in this example *place* and *food*, blue) and instances of attributes (*concepts*, beige). Episodes (sets that contain one concept per attribute), are sampled following a pre-defined probability distribution that depends on each possible pair (**center**, brown). Finally (**right**), the sampled episode is mapped to an input pattern by activating subsets of neurons corresponding to each of the concepts present in the episode. Some final variability within episodes is induced by randomly flipping the activities of $N_{\text{swap}}/2$ active and $N_{\text{swap}}/2$ inactive neurons. **D:** Example of the semantic structure of *Italy* induced by the EGP in 1C. From the probabilities of each possible episode, one can derive the conditional probabilities associated to *Italy* being present in the episode. **E:** Semantic structure of the EGP in 1C.

We find that the semantic structure of the EGP is moderately to highly correlated with recurrent connectivity after learning (Fig. 2C), suggesting the semantic structure, as here defined, has a non-negligible impact in learning for a variety of architectures. Interestingly, the feed-forward network shows stronger alignment at early stages of training, to then move to an identity representation (see Appendix B). The Hopfield-Tsodyks network shows the strongest alignment with the semantic structure, but is inherently limited by its necessarily symmetric connectivity. These results open the door to the possibility of a network extracting semantics in full.

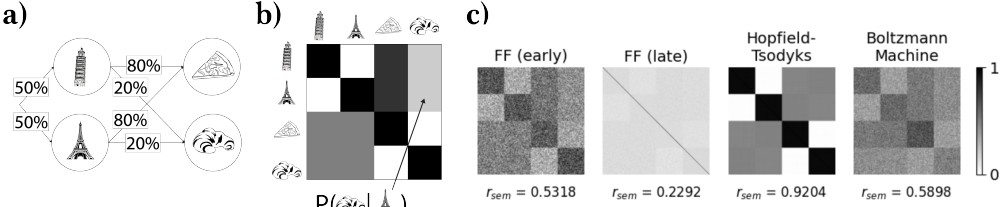

Figure 2: **A:** Diagram summarizing the episode sampling process of this EGP. **B:** Semantic Structure associated to this EGP. **C:** Normalized connectivity for 4 different networks: A feed-forward network trained with self-supervised learning (*early* and *late* stages of learning), a network with Hopfield-Tsodyks connectivity, and a Boltzmann machine with only visible units.

### 2.3 HOMEOSTATIC BINARY NETWORKS (HBNS)

Building on the previous results, we propose a biologically-constrained recurrent neural network (RNN) designed to extract episode semantics without the need for symmetric connections: the **Homeostatic Binary Network** (HBN). HBNs are inspired by classic models of Hebbian learning (Hopfield, 1982; Tsodyks & Feigel'man, 1988) and competitive learning (Rumelhart & Zipser, 1985). Additional homeostatic mechanisms, that here take the form of a generalization of winner-take-all dynamics (using a top-$K$ operation) and synaptic renormalization, are crucially included to facilitate semantic learning. These homeostatic principles align with regularization methods in artificial neural networks (Hofmann & Mäder, 2021), which constrain network activity and weight distribution.

**Network Architecture** The network consists of two distinct regions (*place* and *food*; Fig. 3A), corresponding to inputs representing different attributes. HBNs operate in two modes (Fig. 3A):

- **Input-Driven Mode:** Input determines the activity, ignoring recurrent weights (equation 3).
- **Pattern Completion Mode:** Recurrent weights drive network activity to recall previously learned patterns (equation 4 and equation 10)

**Activity-Level Homeostasis.** HBNs ensure homeostasis at the activity level through binary activation and fixed sparsity constraints. After both input and recurrent processing (Fig. 3A; Eq. equation 3 and equation 4), the network applies a top-$K$ activation function (equation 5). This operation activates the $K$ highest values in each region while suppressing the rest (Fig. 3B), mimicking the effect of adjustable inhibition observed in biological neural networks (Table 1, equation 6).

**Synaptic-Level Homeostasis.** At the synaptic level, homeostasis is achieved through synaptic renormalization (Fig. 3C). This mechanism constrains the sum of incoming and outgoing synaptic weights for each neuron to remain below fixed thresholds ($w_{max}^{in}$ and $w_{max}^{out}$, respectively; Table 1, equation 8 and equation 9). These constraints give rise to three distinct learning regimes, which interact differently with the semantic structure of episodes.

**Weight trajectories and Semantic Structure.** In Appendix C, we show how weight trajectories depend on input statistics. We find that the network primarily can be described by 3 regimes:

- *HBN (out)* : Outgoing homeostasis dominates ($w_{out}^{max} << w_{in}^{max}$)

$$w_{ij} \longrightarrow p(\boldsymbol{x}_i = 1 | \boldsymbol{x}_j = 1) \tag{1}$$

- *HBN (balanced)* : Outgoing and incoming homeostasis are balanced ($w_{out}^{max} \approx w_{in}^{max}$, no analytical solution found)

- *HBN (in)* : Incoming homeostasis dominates ($w_{out}^{max} >> w_{in}^{max}$)

$$w_{ij} \longrightarrow p(\boldsymbol{x}_j = 1 | \boldsymbol{x}_i = 1) \tag{2}$$

where $p(\boldsymbol{x}_i = 1 | \boldsymbol{x}_j = 1)$ is the probability that (during learning), neuron $i$ is active if neuron $j$ is active.

It should be highlighted that synaptic renormalization plays a key role in obtaining the previous results. As pure hebbian learning keeps track of the joint firing probability, homeostatic mechanisms that multiplicatively depresses synapses whenever there is a pre (incoming homeostasis) or a post (outgoing homeostasis) firing transforms a joint distribution into a conditional probability. Furthermore, these results showcase the motivation for HBNs as a model for semantic learning, as synapses converge to conditional firing probabilities, matching the associated semantic structure under the assumption that neurons track input statistics sufficiently well.

**Semantic Learning in Recurrent Connections.** To investigate the representations emerging in HBN's recurrent connections during acquisition, we trained it on the same EGP of the previous section Fig. 2A). Weights from pre-neuron $j$ to post-neuron $i$ were color-coded based on the episode concept each neuron represented (Fig. 3C, right). Synapses were grouped by their conceptual pairings: self-connections (black), within-attribute connections (blue), *place*-to-*pizza* (green), *place*-to-*croissant* (red), and *food*-to-*place* (orange). These groupings reflect shared conditional likelihoods (compare Fig. 3B and Fig. 3C, right).

Weight trajectories were analyzed for three homeostatic regimes: **Outgoing Dominance** ($w_{\max}^{\text{out}} \ll w_{\max}^{\text{in}}$), **Balanced** ($w_{\max}^{\text{in}} = w_{\max}^{\text{out}}$), and **Incoming Dominance** ($w_{\max}^{\text{out}} \gg w_{\max}^{\text{in}}$; Fig. 3D1, right). These regimes were chosen due to a phase transition in learning dynamics (Appendix C; Fig. 3D3), where the outcome largely depends on the homeostatic balance. In this sense, in Fig. 3D3 one can see how the final weights are independent of the out/in homeostasis ratio except for a narrow region (these related to the 3 regimes explained above). Theoretical learning trajectories (dashed lines) closely approximated mean-field simulations (solid lines; Fig. 3D1).

Post-convergence, weights under outgoing and incoming dominance were proportional to conditional firing probabilities, linking them to episodic semantics (Appendix C). Simulations confirmed these results, revealing that the final weight matrix had an associated semantic correlation much higher than previously tested networks (Fig. 3D2). Specifically, connections aligned with the semantic structure for outgoing dominance ($w_{ij}$ reflecting the probability of concept associated to $i$ given concept associated to $j$. For incoming dominance, this was inverted, aligning with the transpose of the semantic structure.

## 2.4 Behavioral Signatures of Semantic Learning

Ultimately, one of the goals of HBNs and other networks studied here is to serve as models of the nervous system. While we have shown that HBNs can encode semantic structures in their connections with a high degree of fidelity, this does not necessarily mean they mimic biological neural networks. In this section, we study the behavioural signatures of HBNs and alternative models previously used to describe learning in humans. To do so, we train different networks using our EGP, and then study network responses in two different tasks: (i) pattern completion of noisy input and (ii) pattern completion of semantically-masked input. In addition to the previous models, we also include here Modern Hopfield network (Ramsauer et al., 2020), which we did not study in section 2.2 due to the absence of explicit recurrent connections. In this sense, the results presented here could be extended to any network that has been trained to recover/generate input patterns.

One motivation of this section is characterizing how different types of network make predictions over out-of-distribution input. But, more importantly, this section also provides with a battery of model-specific predictions that could be tested using a similar protocol in human or animal subjects, for example using the matrix heatmap representation of input vectors as visual cues.

### 2.4.1 Recall of Semantic Prototypes from Noisy Input

It has recently been shown Kang & Toyoizumi (2024) that recurrent neural networks using a Hopfield-Tsodyks connectivity can identify *semantic prototypes* from input. Semantic prototypes can be understood as an average pattern common across many episodes that share a concept, and here would correspond to the list of One-Hot encodings that represent episodic content before adding noise (Fig. 4A-B).

Table 1: HBN Function, Training, and Testing Implementation

| HBN Function and Implementation | | |
|---|---|---|
| **Function** | **Implementation** | **Equation(s)** |
| Input processing | Pre-activation $\mathbf{z}_i$ corresponds to the input $\mathbf{x}_i^{\text{input}}$ | $$z_i = x_i^{\text{input}} \qquad (3)$$ |
| Recurrent processing | Pre-activation for each region $\mathbf{z}_i$ corresponds to the sum of weights $w_{ij}$ times input $\mathbf{x}_j$ | $$z_i = \sum_j w_{ij} x_j \qquad (4)$$ |
| Nonlinear activation | Activation function: $\mathbf{x} = \text{top-}K(\mathbf{z})$. The $K$ neurons in region $l$ with highest $z_i$ have $x_i = 1$ and the rest $x_i = 0$. | $$x_i^{\text{region}} = [\text{top-}K(z^{\text{region}})]_i \qquad (5)$$ |
| Adjustable inhibition | This activation function can be interpreted as a step with a threshold that depends on the layer input $z$ and $K$. | $$x_i = H\Big(z_i - \theta(z; K)\Big) \quad (6)$$ |
| HBN During *Train* (Acquisition) | | |
| **Function** | **Implementation** | **Equation(s)** |
| Hebbian learning | Pre-post pairing. | $$\Delta w_{ij}^{\text{Hebb}} = \lambda \mathbf{x}_i \mathbf{x}_j \qquad (7)$$ |
| Homeostatic plasticity (in) | Multiplicative synaptic renormalization over incoming synapses. When the total sum of incoming weights at post-neuron $i$ exceeds a threshold $w_{\max}^{\text{in}}$ by a value $\epsilon_i$, each weight $w_{il}$ is normalized to impose $\sum_l w_{il} = w_{\max}^{\text{in}}$. | $$\text{if } \sum_l w_{il}^{(t)} = w_{\max}^{\text{in}} + \epsilon_i$$ $$\implies w_{il}^{(t+1)} = w_{il}^{(t)} \frac{w_{\max}^{\text{in}}}{w_{\max}^{\text{in}} + \epsilon_i} \qquad (8)$$ |
| Homeostatic plasticity (out) | Multiplicative synaptic renormalization over outgoing synapses. When the total sum of outgoing weights at pre-neuron $j$ exceeds a threshold $w_{\max}^{\text{out}}$ by a value $\epsilon_j$, each weight $w_{kj}$ is normalized to impose $\sum_k w_{kj} = w_{\max}^{\text{out}}$. | $$\text{if } \sum_k w_{kj}^{(t)} = w_{\max}^{\text{out}} + \epsilon_j$$ $$\implies w_{kj}^{(t+1)} = w_{kj}^{(t)} \frac{w_{\max}^{\text{out}}}{w_{\max}^{\text{out}} + \epsilon_j} \qquad (9)$$ |
| HBN During *Test* (Recall) | | |
| **Function** | **Implementation** | **Equation(s)** |
| Pattern completion | The recurrent layer can project a network state $x$ into its recurrent connections. | $$x^{\text{region}} \leftarrow \text{top-}K\Big([W \cdot x]^{\text{region}}\Big) \qquad (10)$$ |

To test the ability of different networks to recover semantic prototypes from noisy versions of input, we train each network on input with the same amount of noise as in previous section ($N_{\text{swap}} = 4$), and then test each network by providing inputs with varying levels of noise, (from $N_{\text{swap}} = 4$ to $N_{\text{swap}} = 100$, which would correspond to a totally random pattern.

We find that almost all networks tend to drive their neural activity toward the *semantic prototype* (the input pattern before the swaps; Fig. 4B, bottom) of each episode element (Appendix E) and Fig. 4C). Notably, the performance (measured as the cosine similarity between the recovered pattern and the input pattern before noise), follows a very similar curve for all networks except for an HBN with outgoing homeostasis and the Boltzmann machine. In the case of HBN (out), the average performance is comparable to the rest of networks, but it shows a much higher variance. Further inspection shows the reason is that this network is biased to recovering recovering overall-most-likely episodes, even when the original input corresponded to a less likely example. On the other hand, the Boltzmann machine shows an overall lower performance, being the only network that is not able to accurately recover semantic prototypes from input examples.

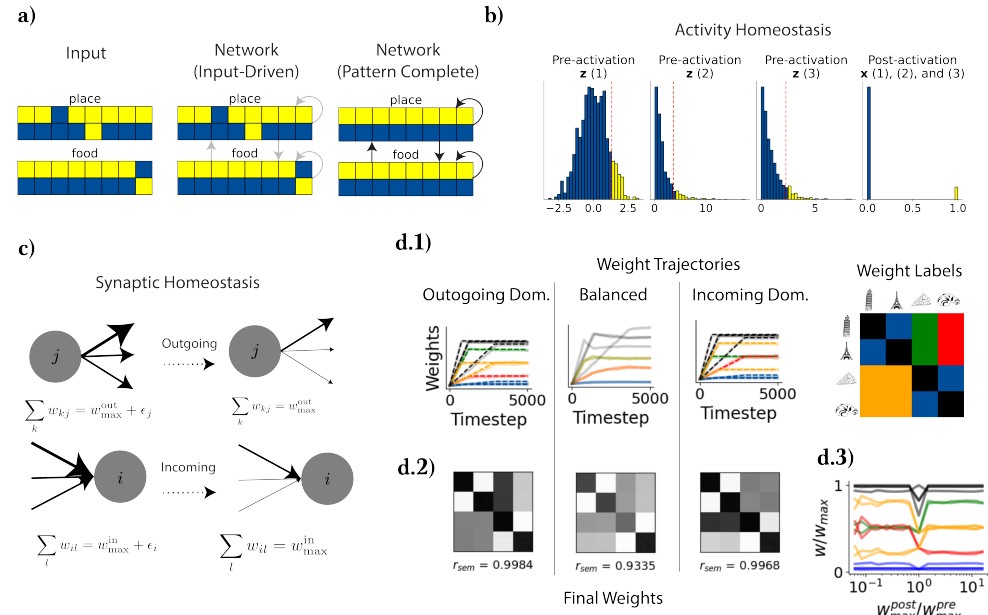

Figure 3: **A:** When Input-Driven, the network activity is the result of performing the top-$K$ activation over the input. During testing, the network can pattern-complete the original input by activating the projection of input-driven activity onto the recurrent weights. **B:** Homeostasis in the network activity imposes a fixed level of sparsity by applying a top-$K$ activation over the pre-activation values. For 3 different pre-activation distributions $z$ (1), (2), and (3) (illustrative, not actual input sampled from our EGP), the post-activation distribution is always $N - K$ 0's and $K$ 1's. **C:** Synaptic homeostasis is implemented via outgoing (**top**) and incoming (*bottom*) homeostasis, which respectively ensure that the total amount of incoming and outgoing weights are bounded to maximum values $w_{\max}^{\text{in}}$ and $w_{\max}^{\text{out}}$. **D:** Semantic Learning in HBNs. **D1:** Weight trajectories across learning, synapses coloured as shown in right. Dashed lines indicate theory (only Outgoing and Incoming Dom.) **D2:** Normalized connectivity and semantic correlation (below), for each network, after learning. **D3:** Final weights for different ratios $w_{\text{out}}^{\max}/w_{\text{in}}^{\max}$.

These results hilight the first behavioural signatures of two specific types of network: HBN (out) can accurately learn semantic prototypes, but is biased to most-likely episodes for very high levels of noise. In contrast, a Boltzmann machine (without hidden units) has a poor performance in prototype learning regardless of noise levels. These two properties could be well captured by behavioural experiments in humans or other animals.

### 2.4.2 PREDICTIVE BIASES IN SEMANTIC LEARNING

Given an input that has been conceptually masked (activity in the region coding for one of the attributes *place* or *food*), one can have different predictive strategies. If a clear cue for *place* is given, what *food* should be predicted? One option is assume the cue given as a prior, and perform Maximum A Posteriori Estimation (MAP) over the different foods:

$$\arg\max_{target} p\Big(target \text{ in episode}|cue \text{ in episode}\Big) \tag{11}$$

In this case, the predicted food is the most likely to appear in an episode, given the cue shown is also present. A different strategy would be predicting the concept that would make the given cue most likely, which would correspond to Maximum Likelihood Estimation (MLE):

$$\arg\max_{target} p\Big(cue \text{ in episode}|target \text{ in episode}\Big) \tag{12}$$

In order to understand how HBNs and the rest of the networks studied relate to these prediction strategies, we train different models on input with a semantic structure as 9. This semantic structure

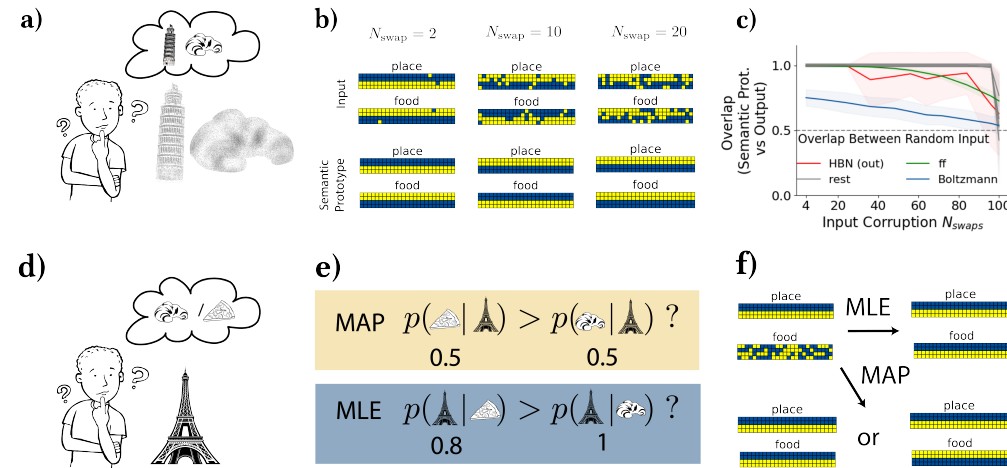

Figure 4: Behavioural signatures of semantic learning in neural networks **A:** Prototype learning, a noisy version of input is presented, and the network has to recover the semantic prototype. **B:** Examples of corrupted input (top) and their corresponding semantic prototype (bottom) **C:** Performance of prototype learning across networks. Shadowed areas show standard deviations. **D**: Recall from partially-masked input. **E:** Schematic of differences between MAP and MLE. **F:** Examples of predicted neural activity upon MAP and MLE.

has been design to obtain fairly dissimilar responses under MLE and MAP paradigms. This also allows obtaining further behavioural signatures of each different form of learning, making specific predictions on participants performance.

Table 2 summarizes the performance of HBNs trained under different regimes, as well as other baseline models. The MAP and MLE scores where obtained by computing the dot product between the behavioural distribution of selected (highest average) concepts given a cue with the known solution from input statistics. MAP bias and MLE bias take only into account those cue-recall pair in which the answer was different for MAP and MLE.

As expected (Appendx F.2), HBN (out) distinctively outperforms the rest of networks in MAP, as well as showing a perfect bias towards this predictive strategy. Similarly, HBN (both balanced and in) show a very good performance and bias in MLE. The rest of networks in general show intermediate to good performances, but all of them have a strong bias towards one or the other. Notably Boltzmann machines have a very good performance in MAP, despite bad performance in prototype learning.

Overall, distinctive behavioural signatures of different have been obtained to be compared with potential experiments. Notably, the space of all possible semantic structures is by all means not fully explored here, with this a single example of applicability of the method.

Table 2: Performance comparison of MAP and MLE metrics. Metrics obtained over trials of 1000 episodes. Standard deviations show difference between 4 different trials of 1000 episodes each.

| Model | MAP score | MLE score | MAP bias | MLE bias |
|---|---|---|---|---|
| HBN (Out) | $0.9954 \pm 0.007$ | $0.67 \pm 0.03$ | $1 \pm 0$ | $0 \pm 0$ |
| HBN (Balanced) | $0.6805 \pm 0.008$ | $0.9990 \pm 0.04$ | $0.04 \pm 0.01$ | $0.9992 \pm 0.0004$ |
| HBN (In) | $0.681 \pm 0.003$ | $0.9990 \pm 0.0003$ | $0 \pm 0$ | $1 \pm 0$ |
| Hopfield-Tsodyks | $0.731 \pm 0.004$ | $0.983 \pm 0.002$ | $0.18 \pm 0.01$ | $0.982 \pm 0.002$ |
| Modern Hopfield | $0.93 \pm 0$ | $0.63 \pm 0.1$ | $1 \pm 0$ | $0 \pm 0$ |
| Feed-Forward | $0.9357 \pm 0.0005$ | $0.5 \pm 0$ | $1 \pm 0$ | $0 \pm 0$ |
| Boltzmann Machine | $0.993 \pm 0.002$ | $0.731 \pm 0.008$ | $0.993 \pm 0.002$ | $0.09 \pm 0.01$ |

## 3   DISCUSSION

To study the impact of episode semantics in learning (semantic learning), we have proposed an Episode Generation Protocol (EGP) -and a corresponding Semantic Structure. We have tested how different Recurrent Neural Networks (RNNs) are able to extract semantics into its recurrent weights. While there exists a non-negligible correlation between the semantic structure and the learned weights, none of the tested networks were able to extract semantics in full. We have proposed a model designed for this purpose (Homeostatic Binary Networks, HBNs). We show that these outperform previously tested networks, with an almost perfect alignment with the semantic structure of the EGP. Then, we explore behavioural signatures of different models typically used in neuroscience. Results suggest that HBNs, in different regimes of synaptic homeostasis, are biased to perform Maximum A Posteriori (MAP) or Maximum Likelihood Estimation (MLE). Similarly, we propose that our EGP can be used in behavioural experiments with animals or humans to validate/falsify the ability of the different networks to explain learning in biological agents.

Our EGP supposes a step forward in the study of semantic learning in artificial neural networks, generalizing frameworks that assume associations between input and output (McClelland & Rogers, 2003; Saxe et al., 2019) and also moving beyond defining semantics as a mere segregation of concepts (Ben-Shaul et al., 2023), which ignore the semantic web that these concepts form. In this sense, we hope our EGP and Semantic Structure inspire future experiments in interpretability in artificial intelligence , providing a framework to systematically study how neural networks capture input statistics in its internal structure. While not in the main scope of this study, an example of the interplay between learning via back-propagation of errors and semantic learning has been shown in **??**. In this sense, we have shown how the semantic structure guides learning trajectories in initial learning phases even in a simple setup where a feed-forward network is trained to be an identity. Crucially, when this training was performed over masked input, the correlation between network and semantic structure was kept consistently high. This result aligns with recent studies reporting *facts* in Large Language Models being stored in the feed-forward connections of its transformer blocks (Nanda et al., 2023), considering that next-word prediction can be seen as a formed of masked self-supervised learning.

HBNs build upon long-standing ideas like Hopfield networks (Hopfield, 1982) and competitive learning (Rumelhart & Zipser, 1985). One relatively novel aspect in HBNs is incorporating outgoing homeostasis, which has also recently been included in Fung & Fukai (2023). While their work focuses on learning in feed-forward connections, competition for presynaptic resources (outgoing homeostasis) was already found to help learning with overlapping patterns.

The learning dynamics in HBNs are heavily impacted by the Semantic Structure of episodes, with final weights actually matching the Semantic Structure matrix when outgoing homeostasis dominates, and its transpose when incoming homeostasis dominates. While their performance in prototype learning is comparable to that of classic models of sparse storage, as can be the Hopfield-Tsodyks (Tsodyks & Feigel'man, 1988) connectivity, HBNs can also capture asymmetric semantic relationships. Furthermore, our results challenge the view that patterns with an overlap over a certain threshold become indistinguishable (Gastaldi et al., 2021). This is crucially allowed by adjustable synaptic inhibition (via our top-$K$ activation function), without which one would get representational collapse. Our study has implications in the reconciliation of error-free (Hebbian) and error-driven (predictive) learning (Kumar, 2021; Zheng et al., 2022). In this sense, although our network model is essentially Hebbian and does not use an explicit error signal, it is able to find asymmetric predictive relations among its input and goes beyond purely associative learning, as shown by its ability to perform Maximum A Priori Estimation (MAP) and Maximum Likelihood Estimation (MLE).

Our work can be placed in the context of Complementary Learning Systems (CLS) theory (McClelland et al., 1995; O'Reilly et al., 2014), via both our Episode Generation Protocol and Homeostatic Binary Networks. Our EGP can be used to formalize the semantic structure of standard input generation protocols in the literature, adding one level of complexity to quasi-orthogonal inputs typically used and adding the notion of semantic structure to those that already use highly overlapping input patterns. As computational models of biological neural networks, HBNs, on the other side, present several advantages over other models of semantization and/or systems consolidation: (i) HBNs are analytically tractable, (ii) HBNs do not use less biologically-plausible

learning mechanisms such as back-propagation (Saxe et al., 2019) or Contrastive Hebbian Learning (Singh et al., 2022), (iii) HBNs are computationally very inexpensive, which strikes compared to other biologically-inspired architectures, many times using spiking networks (Remme et al., 2021; Tomé et al., 2022; Chrysanthidis et al., 2022).

Our work also presents several limitations. In the case of the EGP, the episode-input mapping proposed here is, besides the variability introduced by neuronal activity swapping, the simplest one could think of, making it essentially a one-hot encoding. Future work could study what representations emerge in more complex episode-input mappings, for example introducing a linear-nonlinear relationship between episode concepts and input patterns. Another limitation here, where simplicity in the input has been prioritized in favour of a mechanistic yet intuitive study, is the absence of application to more complex problems to study learning in modern artificial neural networks. Future work could leverage on this limitation by exploring EGPs based on standard benchmarks. An example of this would be imposing a specific semantic structure between pairs of MNIST digits and then presenting a network with stacked images that follow these statistics. While self-supervised learning (Bromley et al., 1993; Gui et al., 2024) has been shown to project input into a latent manifold that is semantics-aware (Ben-Shaul et al., 2023), future work could explore if it also captures semantic relationships between concepts (in this case right and left digit). Finally, another limitation of the EGP in the form presented here, as a model of episodic input generation, is that it does not take into account temporal correlations present in real-life episodes. Extending the protocol in this sense could lead to better understanding how temporal aspects of semantics can be extracted in neural networks.

HBNs, while presenting multiple advantages, crucially depend on parameters such as $K$ that here where imposed to match known input structure and statistics. Understanding how $K$ (which is equivalent to the sparsity levels) can be meta-learned to optimize learning would be primordial to asses its applicability both in AI and neuroscience. Additionally, the model and the theoretical derivations included make several assumptions that also challenge its validity, such as the ability of the network to perfectly track input statistics and disconnect from recurrent weights during learning. While these are standard practices in computational models of biological learning (Clark & Abbott, 2024), activity and learning dynamics are not perfectly decoupled in actual neural systems. Finally, connected to a limitation of the EGP previously highlighted, learning in HBNs does not take into account temporal sequences. The ability to make predictions in time is crucial for episodic and semantic memory, and extensions of the model to allow this, as done in Chaudhry et al. (2024) would be crucial to obtain a model that aims at fully capture learning in biological agents.

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

## A   GENERATING OVERLAPPING INPUT PATTERNS WITH AN EPISODE GENERATION PROTOCOL

### A.1   EPISODE GENERATION PROTOCOL

The Episode Generation Protocol (EGP) is a generative process that samples input neural activity. Our proposed EGP is defined as follows:

#### EPISODES, EPISODE ATTRIBUTES AND EPISODE CONCEPTS

All *episodes* share a structure, such that every episode contains one *episode concept* per *episode attribute*. In this sense, if attributes are $A, B, C...$, an episode is constructed by selecting a concept $a \in A$, another $b \in B$, another $c \in C$... The set of all possible episodes can thus be defined as:

$$E = \{\{a, b, c, \dots\} \mid a \in A, b \in B, c \in C, \dots\} \tag{13}$$

In other words, an episode $e \in E$ is a collection of episode concepts $a$, $b$, $c$, ... such that concept $a$ belongs to episode attribute $A$, concept $b$ belongs to episode attribute $B$, etc.

In this study, for illustrative purposes, we assume that episodes are related to lunch experiences, such that one attribute is *place* and another attribute is *food*. Episodes are therefore specified by choosing $p_i \in place$ and $f_j \in food$, resulting in episodes of the form

$$E = \{\{p_i, f_j\} \mid p_i \in place, f_j \in food\} \tag{14}$$

#### EPISODE PROBABILITY DISTRIBUTION

As a generative process, our EGP samples episodes as a previous step to sampling an input pattern. This is done by fixing a probability distribution over episodes. In our particular example of *place* and *food*, one has to fix

$$P\big(e = \{p_i, f_j\}\big) \tag{15}$$

such that

$$0 \leq P\big(e = \{p_i, f_j\}\big) \leq 1 \ , \ \sum_{i,j} P(e = \{p_i, f_j\}) = 1 \tag{16}$$

#### EPISODE-INPUT MAPPING

After an episode $(p_i, f_j)$ has been sampled, the EGP returns an input pattern. Here, we assume the input vector can be split into a *place* and a *food* region. Then, for each region, we use a list of $K$ repeated One-Hot encodings, where the encoding represents which of all the concepts corresponding to that attribute is present in the episode. Intuitively, each region corresponds to a single attribute, and can be visualized as a matrix that contains as many rows as possible episode concepts in that attribute, and $K$ columns. All entries of the row that correspond to the concept present in the episode are 1 and the rest are 0.

To account for variability between different presentations of the same episode (not all episodes, even though they contain the same concepts, will be exactly the same), we further add some stochasticity by randomly picking $N_{swap}/2$ inactive neurons and $N_{swap}/2$ active neurons and inverting their activity (a total of $N_{swap}$ neurons randomly change their activity). This ensures that activity sparsity in the sensory layer is maintained (the number of flips from 0 to 1 is the same as the number of flips from 1 to 0).

### A.2   SEMANTIC STRUCTURE OF AN EGP

We refer to *semantic field theory* (Bussmann et al., 2006) in order to define what is the *semantic structure* of an EGP. According to this school, the meaning of a word is not isolated but dependent on its relation to the rest of the words. While our task is not one of language, we can use this same paradigm to define the *meaning* of episode concepts. In this sense, the meaning of our episode concepts depends on how they are related to the rest. Intuitively, even if a house is exactly the same for two dogs, the meaning of that house for each will be very different if it

is always presented with food to one dog and always presented with an annoying whistle to the other.

In this light, we use the conditional probabilities of being present in an episode between episode concepts

$$P(i \in e | j \in e) = \frac{P(i \in e, j \in e)}{P(j \in e)} \ \ \forall i, j \in A \cup B \cup C \cup ... \tag{17}$$

as a proxy for the *semantic structure* of an EGP, which we define as the matrix of these conditional probabilities:

$$\text{semantic structure} \equiv \{SS_{ij}\} \ ; SS_{ij} = P(i \in e | j \in e) \tag{18}$$

In other words, extracting the semantics of an EGP is equivalent to: (i) identifying episode concepts, and (ii) extracting how likely is one episode concept $i$ to be present in an episode $e$ if an episode concept $j$ is also present.

One detail that should be noted is that episode attributes are simply groups of concepts the sub-matrices of which are diagonal (they never co-occur together), and one does not necessarily have to define them explicitly (as we do with *place* and *food*), nor they have to exist for that matter (it could happen that all pairs of concepts have a non-zero probability of co-occurrence, so each concept is its own episode attribute). However, here we use the notion of episode attributes to obtain simpler semantic structures the intuition of which can be grasped more easily (by enforcing many zeros in the matrix $SS$). Thus, while in practice the episode attributes are a property of the semantic structure and do not need to be specified a priori in the EGP, here we do so as a story-telling trick, slightly sacrificing the generality of our EGP definition.

### A.3 EGP AND SEMANTIC STRUCTURE OF INPUT USED IN PREVIOUS STUDIES

#### A.3.1 OVERLAPPING PATTERNS FROM SCHAPIRO ET AL. (2017B); SINGH ET AL. (2022)

A very interesting previously used protocol of generation of overlapping input is that used in Singh et al. (2022), in turn inspired in an earlier experimental study (Schapiro et al., 2017b). In these studies, inputs are drawings of *satellites*, together with textual attributes such as their name or class (satellites of the same class share many visual features, and the class feature itself). Given that each satellite is defined by: *name*, *class*, and *visual feature* 1 to 5, our protocol can also be used as a framework (Fig. 5A), using an episode-input mapping that similarly encodes each attribute in a separate network using a list of One-Hot encodings.

The associated Semantic Structure (Fig. 5B, left), which can be learned using an HBN with outgoing homeostasis (Fig. 5B, right) uncovers each of the different episode attributes, but also the highly asymmetric relationship that exists between them (Fig. 5B). For instance, while each name conditions with probability one the class (alpha, beta, gamma), classes condition more weakly the name, as given a class there are 4 possible associated names. Representation Similarity Analysis (RSA) (Kriegeskorte et al., 2008) was successfully used in the past to understand semantics in this dataset, which revealed a strong community structure. However, the relationships highlighter by the Semantic Structure proposed here could not be uncovered, as RSA is an inherently symmetric measure.

**A**

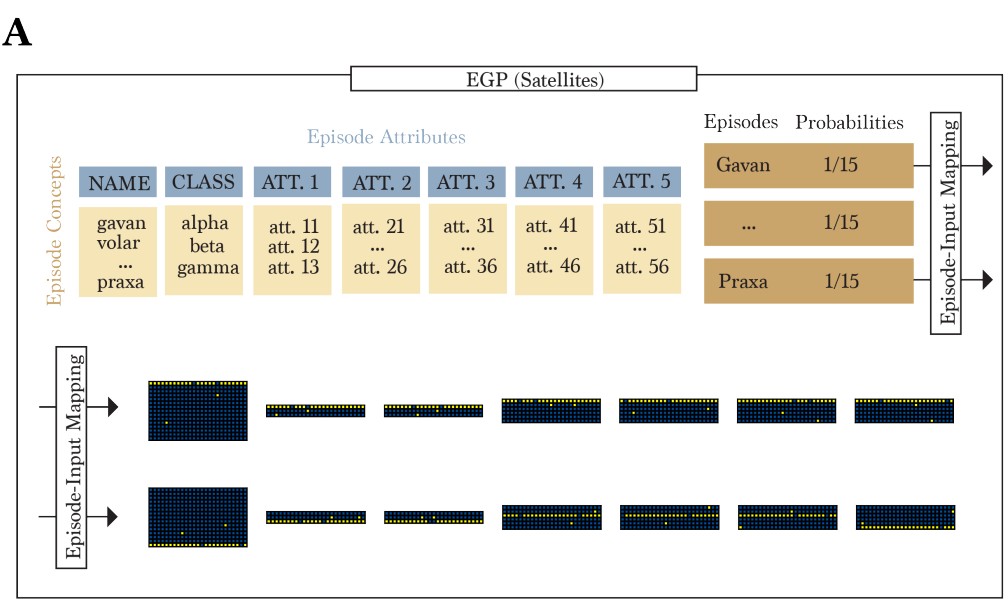

**B**

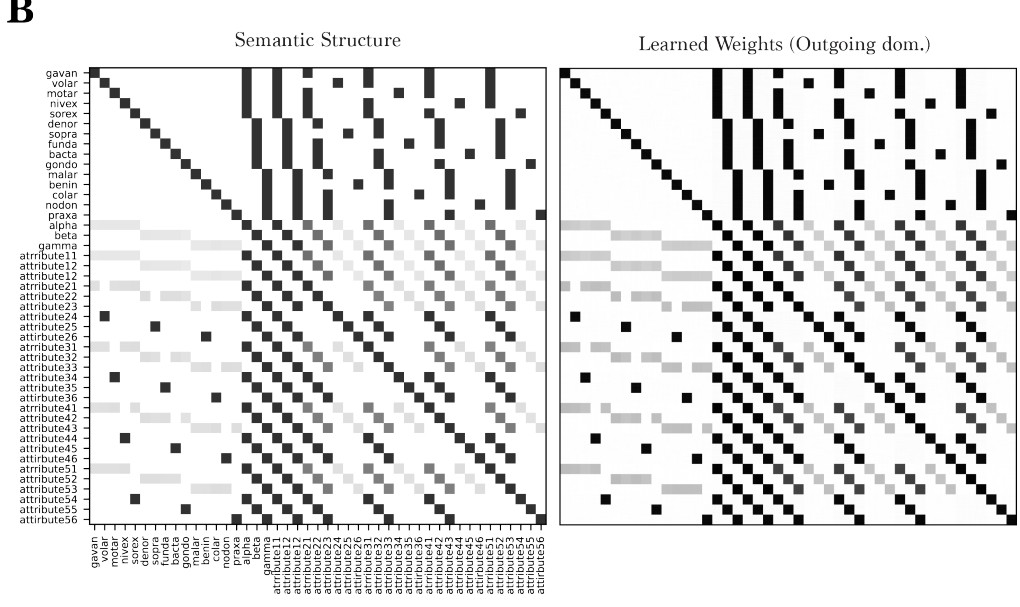

Figure 5: **A:** EGP that mimics input generation in Singh et al. (2022). **B:** Obtained Semantic Structure (left) and weight matrix (right after training with outgoing homeostasis dominance.)

A.3.2 OVERLAPPING PATTERNS FROM FUNG & FUKAI (2023)

To give another example of application in formalizing the generation of overlapping input with our EGP, we also apply it to the input used in Fung & Fukai (2023). This very simple toy input was used to understand how competition on presynaptic resources can aid pattern separation in a feed-forward network. The patterns used were

$$
\begin{aligned}
(0, 1, 1, 0, 0, 0, 0, 0, 0) &\longrightarrow p_1 \\
(0, 0, 1, 1, 0, 0, 0, 0, 0) &\longrightarrow p_2 \\
(0, 0, 0, 0, 0, 1, 1, 0, 0) &\longrightarrow p_3 \\
(0, 0, 0, 0, 0, 0, 1, 1, 0) &\longrightarrow p_4
\end{aligned}
\tag{19}
$$

By noting that positions 2 and 7 are *shared* among patterns $p_1$ and $p_2$, while positions 1,3,6, and 8 are distinctive (*not-shared*) in each of the 4 patterns, our EGP can be used to frame this input generation as shown in Fig. 6A. This yields a semantic structure whereby *not-shared* attributes completely predict (conditional probability of 1) the *shared* attributes, while the opposite is only partially true (conditional probability of 0.5) (Fig. 6B).

**A**

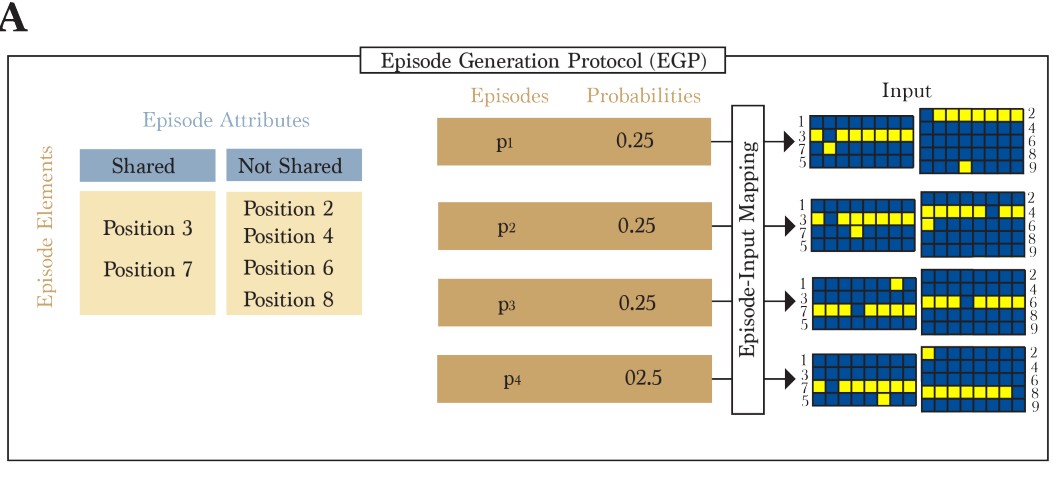

**B**

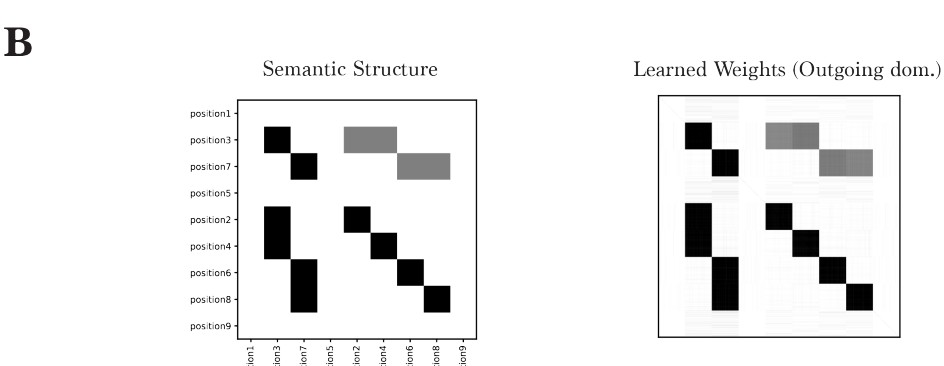

Figure 6: **A:** EGP that mimics input generation in Fung & Fukai (2023). **B:** Obtained Semantic Structure (left) and weight matrix (right after training with outgoing homeostasis dominance).

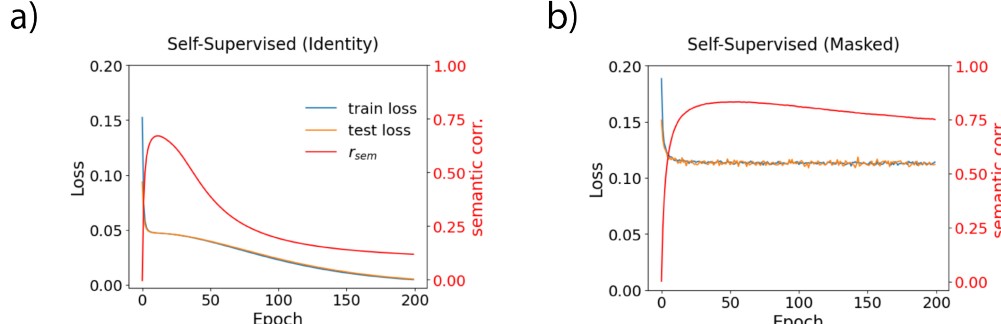

Figure 7: Semantic learning in a feed-forward network. **A:** Loss and semantic correlation across training, identically using the input as output. **B:** Same as B, but masking one of the two regions.

## B   FEED-FORWARD NETWORK

A step in recurrent processing (equation 10) can be seen as the output of a feed-forward network of the same input and output size. To study semantic learning in a setup more closely related to artificial neural networks, we train a neural network consisting of an $N \times N$ linear mapping and a sigmoid. In section 2.2, we train a feed-forward network to simply output the input (note how the solution is trivially an identity matrix). Interestingly, learning dynamics are first guided by semantic learning, with semantic correlation initially increasing (*early* in 2.2), to then slowly shifting towards the actual solution (*late* in 2.2).

## C  MEAN-FIELD LEARNING DYNAMICS (NO PHYSICAL CONNECTIONS)

Here we derive the learning dynamics of a weight $w_{ij}$. The intuition is very similar to that presented in Rumelhart & Zipser (1985), which establishes the fixed points of a feed-forward network with incoming homeostasis.

By including Hebbian, outgoing and incoming homeostatic plasticity, the weight change $\Delta w_{ij}$ at each timestep $t$ is given by:

$$\Delta w_{ij} = \lambda x_i x_j - H\Big(S_i^{\text{out}}(t) - w_{\max}^{\text{out}}\Big)\Delta w_{ij}^{\text{out}} - H\Big(S_i^{\text{in}}(t) - w_{\max}^{\text{in}}\Big)\Delta w_{ij}^{\text{in}} \tag{20}$$

where we have defined the dynamic variables

$$S_j^{\text{out}}(t) = \sum_k w_{kj} \ ; \ \ S_i^{\text{in}}(t) = \sum_l w_{il} \tag{21}$$

which denote the total amount of presynaptic (outgoing) $S_j^{\text{out}}$ and postsynaptic (incoming) $S_i^{\text{in}}$ connectivity. Note how every neuron in the network has associated these two variables. $H(x)$ is the Heaviside function, which takes value 1 if $x \geq 0$ and 0 otherwise. This ensures that outgoing (incoming) homeostatic plasticity only depresses synapses if the total outgoing (incoming) connectivity is above $w_{\max}^{\text{out}}$ ($w_{\max}^{\text{in}}$).

With this formulation, we now aim to obtain the mean-field dynamics (average weight change of a synapse). In this derivation, we will assume the neuronal firing probability distributions $p(x_i = 1, x_j = 1)$, $p(x_i = 1)$, and $p(x_j = 1)$ are known and fixed (in particular, these do not depend on the synaptic state, i.e. there are *no physical connections*). Also, to alleviate notation, we will use $p(i, j) \equiv p(x_i = 1, x_j = 1)$, $p(i) \equiv p(x_i = 1)$, and $p(j) \equiv p(x_j = 1)$.

### NO HOMEOSTATIC PLASTICITY REGIME

If a synapse has an initial state $w_{ij}^0$, $S_j^{\text{out}}(t = 0) < w_{\max}^{\text{out}}$, and $S_i^{\text{in}}(t = 0) < w_{\max}^{\text{in}}$, then the mean-field dynamics can be simplified to:

$$\langle w_{ij}(t) \rangle = w_{ij}^0 + \lambda p(i, j) t \tag{22}$$

$$\langle S_j^{\text{out}}(t) \rangle = S_j^{\text{out}}(0) + \lambda K p(j) t \ ; \ \ \langle S_i^{\text{in}}(t) \rangle = S_j^{\text{in}}(0) + \lambda K p(i) t \tag{23}$$

Intuitively, as long as the threshold in total postsynaptic or presynaptic connectivity are not met, there is only hebbian learning, which increases synaptic efficacy proportionally to time and the probability of neurons $i$ and $j$ firing together. Similarly, also due to hebbian plasticity, the total presynaptic and postsynaptic connectivity linearly increase with time, the marginal likelihood of the pre and postsynaptic neurons to fire, and the amount of active neurons at each timestep (which is explicitly controlled to be $K$ via the top-$K$ activation function). The advantage of obtaining the exact temporal evolution of these variables is that from the own equations one can obtain their temporal validity. This can be done by computing the average time it would take the total connectivity variables to reach the threshold imposed by $w_{\max}^{\text{out}}$ and $w_{\max}^{\text{in}}$:

$$T_j^{\text{out}} = \frac{w_{\max}^{\text{out}} - \lambda K p(j)}{S_j^{\text{out}}(0)} \ ; \ \ T_i^{\text{in}} = \frac{w_{\max}^{\text{in}} - \lambda K p(j)}{S_j^{\text{in}}(0)} \tag{24}$$

Resulting in equation 22 and equation 23 being valid if and only if $t < \min(T_j^{\text{pre}}, T_i^{\text{post}})$.

### ONLY OUTGOING HOMEOSTASIS

We will now assume a synapse in which $T_j^{\text{out}} < T_i^{\text{in}}$ (that is, outgoing homeostasis starts taking place before incoming homeostasis). Under these conditions, let's consider a time $t$ such that $T_j^{\text{out}} \leq t \leq T_i^{\text{post}}$. Then, at any time that $T_j^{\text{out}}(t) > w_{\max}^{\text{out}}$, synapse $w_{ij}$ will be depressed following

$$S_j^{\text{out}}(t) > w_{\max}^{\text{out}} \implies w_{ij}^{t+1} = w_{ij}^t \frac{w_{\max}^{\text{out}}}{S_j^{\text{out}}(t)} \iff \Delta w_{ij}^t = w_{ij}^t \Big[ \frac{w_{\max}^{\text{out}}}{S_j^{\text{out}}(t)} - 1 \Big] \tag{25}$$

In our model, the only source of potentiation is hebbian learning. Therefore, if $S_j^{\text{out}}(t) = w_{\text{max}}^{\text{out}}$ at a given time, the condition $S_j^{\text{out}}(t) > w_{\text{max}}^{\text{out}}$ can only be met upon future firing of neuron $j$. Furthermore, synaptic renormalization (which guarantees that $\sum_k (w_{kj}^t + \Delta w_{ij}^t) = w_{\text{max}}^{\text{out}}$), would immediately drive $S_j^{\text{out}}(t)$ back to $w_{\text{max}}^{\text{out}}$. Eq. equation 25 can be expanded by rewriting $S_j^{\text{out}}$ as $w_{\text{max}}^{\text{out}} + \epsilon_j^{\text{out}}$ and then using $\frac{1}{1+\epsilon} \approx 1 - \epsilon$:

$$\Delta w_{ij}^{\text{in}}(t) = w_{ij}^t \left[ \frac{1}{1 + \epsilon^{\text{pre}}/w_{\text{max}}^{\text{out}}} - 1 \right] = -w_{ij}^t \frac{\epsilon_j^{\text{out}}}{w_{\text{max}}^{\text{out}}} \tag{26}$$

Note how $\epsilon_j^{\text{out}}$ corresponds to the amount of extra presynaptic connectivity of neuron $j$ with respect to the threshold $w_{\text{max}}^{\text{out}}$. This quantity can be obtained by taking into account that (i) homeostatic plasticity consistently resets the total sum to $w_{\text{max}}^{\text{out}}$ and (ii) any extra connectivity has to come from the connections formed with a single presynaptic firing, which leads to an increase of $K$ in $S_j^{\text{out}}$. Then, the average synaptic change over time can be expressed as:

$$\langle \Delta w_{ij}^t \rangle = \lambda p(i, j) - w_{ij}^t \frac{\lambda K p(j)}{w_{\text{max}}^{\text{out}}} \tag{27}$$

which can be interpreted as an exponential decay in $w_{ij}$:

$$d\frac{\langle w_{ij} \rangle}{dt} = -\frac{1}{\tau_w} \left( \langle w_{ij} \rangle - w_{ij}^{\text{out}}(\infty) \right) \tag{28}$$

with

$$\tau_w \equiv \frac{w_{\text{max}}^{\text{out}}}{\lambda K p(j)} \quad ; \quad w_{ij}^{\text{pre}}(\infty) \equiv \frac{w_{\text{max}}^{\text{out}}}{K p(j)} p(i, j) \propto p(i|j) \tag{29}$$

The latter is a result of notable importance, connecting the fixed point (in the absence of incoming homeostasis) with the conditional firing probabilities of neurons $i$ and $j$. The closed-form solution of $\langle w_{ij}(t) \rangle$ is:

$$\langle w_{ij}(t) \rangle = (1 - \beta_w)\langle w_{ij}(T_j^{\text{out}}) \rangle + \beta w_{ij}^{\text{out}}(\infty) \quad ; \quad \beta_w \equiv 1 - \exp\left( -(t - T_j^{\text{out}})/\tau_w \right) \tag{30}$$

, with $\langle S_j^{\text{out}}(t) \rangle$ remaining constant in time:

$$\langle S_j^{\text{out}}(t) \rangle = w_{\text{max}}^{\text{out}} \tag{31}$$

$\langle S_i^{\text{in}}(t) \rangle$ can be obtained from its definition

$$\langle S_i^{\text{in}}(t) \rangle = \sum_l \langle w_{il}(t) \rangle \tag{32}$$

Using equation can be slightly computationally inefficient, as one first has to obtain all the values $\langle w_{ij}(t) \rangle$ and then check if the assumption that $S_i^{\text{in}}(t) < w_{\text{max}}^{\text{in}}$ is met. That can be bypassed only in the case where all neurons have the same firing rate $p(i) = p(j) = p$, where a simplified expression that does not require computing the weights before knowing the validity of that computation:

$$S_i^{\text{in}}(t) = (1 - \beta_{\text{in}})S_i^{\text{in}}(T_j^{\text{out}}) + \beta_{\text{in}}S_j^{\text{in}}(\infty) \tag{33}$$

with

$$S_j^{\text{in}}(\infty) = w_{\text{max}}^{\text{out}} \quad ; \quad \beta_{\text{in}} \equiv 1 - \exp\left( -(t - T_j^{\text{out}})/\tau_{\text{in}} \right) \quad ; \quad \tau_{\text{in}} \equiv \frac{w_{\text{max}}^{\text{out}}}{\lambda K p} \tag{34}$$

An important case to consider is when $S_j^{\text{in}}(\infty) < w_{\text{max}}^{\text{in}}$, as incoming homeostasis is effectively not present, given that the postsynaptic connectivity is guaranteed to be below the threshold $w_{\text{max}}^{\text{in}}$. This is reflected in the expression for $T_i^{\text{in}}$, which now is:

$$T_i^{\text{in}} = -\tau_{\text{in}} \log\left( \frac{w_{\text{max}}^{\text{in}} - S_j^{\text{in}}(\infty)}{S_i^{\text{in}}(T_j^{\text{in}}) - S_j^{\text{in}}(\infty)} \right) \tag{35}$$

and only takes real values for $S_j^{\text{in}}(\infty) < w_{\text{max}}^{\text{in}}$. If this condition is not met, then the total amount of postsynaptic connectivity of neuron $i$ eventually reaches $w_{\text{max}}^{\text{in}}$ at $t = T_i^{\text{in}}$.

ONLY INCOMING HOMEOSTASIS

A similar approach can be used to obtain the solutions when incoming homeostasis dominates $(T_j^{\text{out}} > T_i^{\text{in}})$.

SCALING COMPETITION FOR PRE- AND POSTSYNAPTIC RESOURCES

In view of the previous results, one can define as $w_{\max}^{\text{out}} \equiv \eta^{\text{out}} K w_{\max}$ (and similarly $w_{\max}^{\text{in}} \equiv \eta^{\text{in}} K w_{\max}$), with $\eta^{\text{in}} \eta^{\text{out}}, \eta^{\text{in}}$ determining the fraction of pre- and postsynaptic resources (respectively) with respect to a value $w_{\max}$, to obtain

$$w_{ij} \begin{cases} = \eta_{\text{out}}\ p(\boldsymbol{x}_i^{\text{post}} = 1 | \boldsymbol{x}_j^{\text{pre}} = 1) w_{\max} & \text{if outgoing homeostasis dominates} \\ = \eta_{\text{in}}\ p(\boldsymbol{x}_j^{\text{pre}} = 1 | \boldsymbol{x}_i^{\text{post}} = 1) w_{\max} & \text{if incoming homeostasis dominates} \end{cases} \tag{36}$$

with $\eta_{\text{out}}/\eta_{\text{in}}$ determining the homeostatic balance of the network. In the following section we find what are the input-driven network statistics, which allow calculating equation 36.

# D  CONNECTION BETWEEN EPISODE GENERATION PROTOCOL AND INPUT-DRIVEN NETWORK STATISTICS

In order to connect our EGP with the learning dynamics established in Appendix C, we find the expression for the marginal $p(i)$, $p(j)$ and joint $p(i, j)$ probabilities of two neurons to fire. In the absence of episode variability, where every neuron's activity and the episode concepts map 1 to 1, $p(i, j)$ corresponds to the probability of the episode whose concepts are coded by neurons $i$ and $j$.

If one takes into account the intra-episode variability, these probabilities are shifted, as every originally active neuron has a probability $N_{\text{swap}}/(2K)$ of being off, and every originally inactive neuron a probability $N_{\text{swap}}/(2(N - K))$ of being on. From this, one can obtain $p(i)$ as a function of the original marginal firing probability $p_0(i)$:

$$p(i) = \frac{K/2 - N_{\text{swap}}/4}{K/2} \frac{K/2 - N_{\text{swap}}/4 - 1}{K/2 - 1} p_0(i) + \frac{N_{\text{swap}}/4}{N/2 - K/2} \big(1 - p_0(i)\big) \tag{37}$$

where we have assumed two regions of size $N/2$ and $K/2$ active neurons in each.

In order to obtain the joint probability of two neurons $i$ and $j$ being active, there are 4 possible scenarios that could lead to such final state:

1. The two neurons are originally on, and none of them flips
2. Only neuron $i$ is originally on, and neuron $j$ flips
3. Only neuron $j$ is originally on, and neuron $i$ flips
4. Both neurons are originally off, and both neurons flip

At the same time, these different scenarios will yield a different result depending on whether the two neurons belong to the same region or not. If they belong to the same region, the probability of no neuron flipping is:

$$p_{(i)} = \frac{\left(\frac{K}{2} - \frac{N_{\text{swap}}}{4}\right)\left(\frac{K}{2} - \frac{N_{\text{swap}}}{4} - 1\right)}{\frac{K}{2}\left(\frac{K}{2} - 1\right)} \tag{38}$$

Given scenarios (ii) and (iii), and given the symmetry between both, the probability of one active neuron not flipping and one inactive neuron flipping is (not that now the two processes are independent):

$$p_{(ii)} = p_{(iii)} = \left(\frac{\frac{K}{2} - \frac{N_{\text{swap}}}{4}}{\frac{K}{2}}\right) \cdot \left(\frac{\frac{N_{\text{swap}}}{4}}{\frac{N}{2} - \frac{K}{2}}\right) \tag{39}$$

Finally, in scenario (iv), the probability that both inactive neurons flip is:

$$p_{(iv)} = \frac{\frac{N_{\text{swap}}}{4}}{\frac{N}{2} - \frac{K}{2}} \cdot \frac{\frac{N_{\text{swap}}}{4} - 1}{\frac{N}{2} - \frac{K}{2} - 1} \tag{40}$$

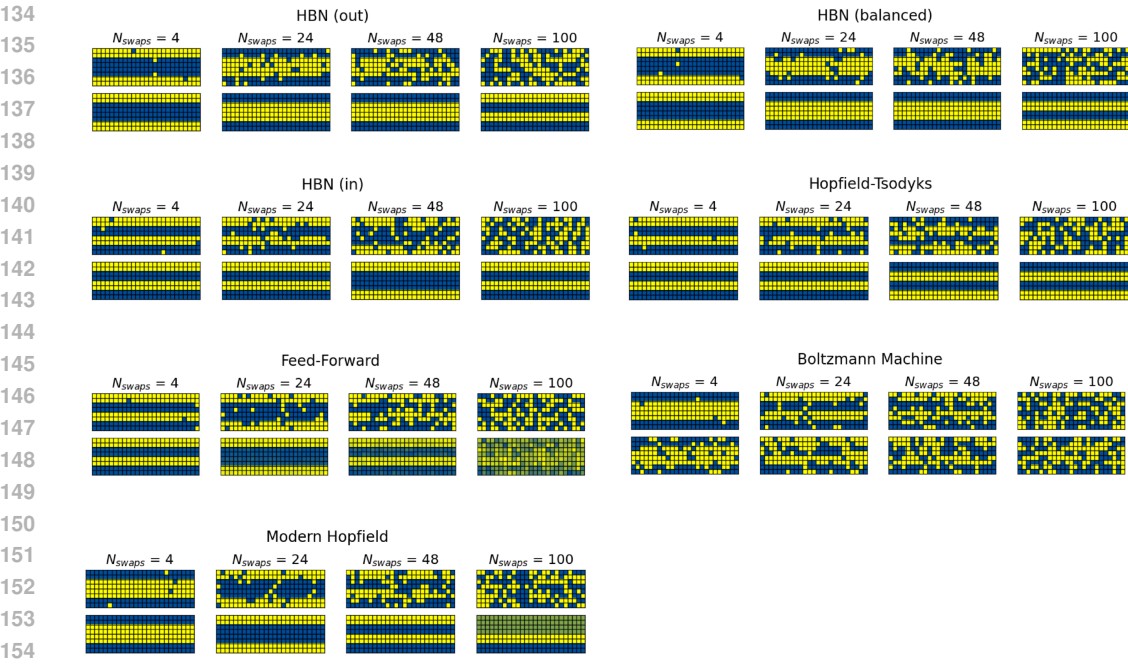

Figure 8: Example output patterns given noisy input across different tested networks

If, on the contrary, the two neurons belong to two different regions, then, for scenario (i), the probability of both neurons being on is;

$$p_{(i)} = \frac{\left(\frac{K}{2} - \frac{N_{\text{swap}}}{4}\right)\left(\frac{K}{2} - \frac{N_{\text{swap}}}{4}\right)}{\frac{K}{2} \cdot \frac{K}{2}} \tag{41}$$

for scenarios (ii) and (iii):

$$p_{(ii/iii)} = \left(\frac{\frac{K}{2} - \frac{N_{\text{swap}}}{4}}{\frac{K}{2}}\right) \cdot \left(\frac{\frac{N_{\text{swap}}}{4}}{\frac{N}{2} - \frac{K}{2}}\right) \tag{42}$$

and for scenario (iv):

$$p_{(iv)} = \frac{\frac{N_{\text{swap}}}{4}}{\frac{N}{2} - \frac{K}{2}} \cdot \frac{\frac{N_{\text{swap}}}{4}}{\frac{N}{2} - \frac{K}{2}} \tag{43}$$

Thus, the total probability of two neurons being active is

$$p(x_i = 1, x_j = 1) = p_{(i)} p_0(x_i = 1, x_j = 1) \; +$$

$$p_{(ii/iii)}\Big(p_0(x_i = 1, x_j = 1) + p_0(x_i = 0, x_j = 1)\Big) \; +$$

$$p_{(iv)} p_0(x_i = 0, x_j = 0) \tag{44}$$

# E  PROTOTYPE LEARNING ACROSS NETWORKS

# F  MAXIMUM A POSTERIORI AND LIKELIHOOD ESTIMATION IN HOMEOSTATIC BINARY NETWORKS

## F.1  SEMANTIC STRUCTURE

In this section, we use a different Semantic Structure with the goal of having key elements that are distinctive (are different in MAP and MLE).

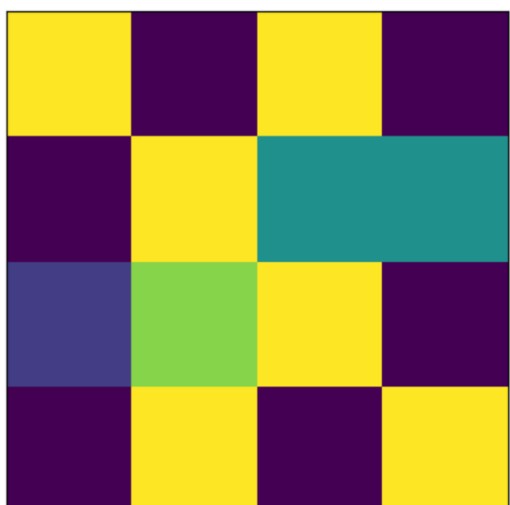

Figure 9: Semantic Structure Used in MLE and MAP estimation

### F.2 Connection with learning in HBNs

Here, we show how, in the limit $N_{\text{swap}} \to 0$, the network follows Maximum A Posteriori Estimation (MAP) and Maximum Likelihood Estimation (MLE) in the limits $w_{\text{out}}^{\text{max}} >> w_{\text{in}}^{\text{max}}$ and $w_{\text{out}}^{\text{max}} << w_{\text{in}}^{\text{max}}$ (respectively). The sketch idea is using the fact that in these regimes weights reflect conditional appearance probability between episode elements. If the input for one region (in this case place) is given, and the rest is random, recurrent (pre-activation) input depends linearly on these conditional probabilities. After applying the top-K operation the element with highest probability is set to 1 and the rest to 0.

We start by introducing the following notation: $x_i^{p/f}$ is the activity of the $i$th neuron coding for either a place concept $p \in P \equiv \{p_1, p_2\}$ or a food concept $f \in F \equiv \{f_1, f_2, f_3, f_4\}$ (each possible place and food). A food-masked vector corresponding to an episode $e = p_e, f_e$ is characterized as

$$x_i^p = \begin{cases} 1 & \text{if } p \in e \\ 0 & \text{else} \end{cases} \tag{45}$$

Note how this reshapes the vectors $x$ from $(N)$ to (regions, attributes per region, neurons per element). Following this notation, an input vector where the attribute *food* is masked, and the episode is $e$, is defined as:

$$x_i^{\text{food},f} = 0 \tag{46}$$

$$x_i^{\text{place},p} = \begin{cases} 1 & \text{if } p \in e \\ 0 & \text{else} \end{cases} \tag{47}$$

that is, an input vector is only 1 in the neurons coding for the place present in episode $e$ and 0 elsewhere.

$$z_i^f = \sum_p \sum_j w_{ij}^{fp} x_j^p + \sum_{f'} \sum_j w_{ij}^{ff'} x_j^{f'} \tag{48}$$

Using that $x_j^p = \delta_{pp_e}$ and that in both regimes $w_{ij}^{ff'} = \delta_{ff'}$, this can be simplified to

$$z_i^f = K w^{fp_e} + \sum_j w_{ij}^{ff} x_j^f \tag{49}$$

### F.3 MAXMIMUM A POSTERIORI ESTIMATION (MAP)

In the limit of outgoing dominance, using EQUATION in equation 50,

$$\boldsymbol{z}_i^f = Kp(f \in e | p_e \in e) + N_f \tag{50}$$

Given that $N_f$ is independent of $f$, and that the top-$K$ operation is independent of an overall bias and scale, assuming deviations in $N_f$ with respect to $K$ is small

$$\boldsymbol{x}_i^f = 1 \iff f = \arg\max_f p(f \in e | p_e \in e) \tag{51}$$

### F.4 MAXIMUM LIKELIHOOD ESTIMATION (MLE)

A similar reasoning can be followed in the limit of incoming homeostatic dominance, to obtain

$$\boldsymbol{x}_i^f = 1 \iff f = \arg\max_f p(p_e \in e | f \in e) \tag{52}$$

