# OpenReview forum: "Predicting episodic structure from overlapping input in binary networks with homeostasis"
_ICLR.cc/2025/Conference — Submitted to ICLR 2025_

### Official Review · Reviewer_nZZ9 · 2024-11-01

**Soundness:** 2
**Presentation:** 1
**Contribution:** 3
**Rating:** 5
**Confidence:** 4

**Summary:**

In their paper “Predicting episodic structure from overlapping input in binary networks with homeostasis” authors develop a new neural network, the Homeostatic Binary Network (HBN) which, through local leaning / updating rules learn to pattern complete noisy inputs. In the context of their investigations, they also develop a dataset that can construct multiple data examples from a fixed semantic relationship, by allowing for the representation of noise on top of class representations. They show that specific implementations of their model follow the principles of either Maximum A Posteriori (MAP) or Maximum Likelihood Estimation (MLE) when completing and unseen pattern.

**Strengths:**

The topic of Hopfield networks and other related architectures have regained in popularity in recent years, and it seems that in principle this paper might make an interesting contribution by introducing a new architecture which matches existing algorithms in performance but also allows for understanding of the networks internal mechanisms used for pattern completion. Additionally, their architecture can be configured to follow different pattern completion strategies.

**Weaknesses:**

I personally find that the presentation of findings and the contextualisation of finding is quite poor in this paper. I provide a detailed list of suggested changes below which I think will help to improve the author’s work.

**Questions:**

I am voting against acceptance for the manuscript in its current form but if the writing and presentation is improved substantially, as suggested below, then I think the results should be worthwhile to be presented at the conference.

*Major: description of their new methods*

When introducing both their dataset and their model, I think it would help the general reader if they described their methodological advancements in context of established works that readers are likely familiar with. For example, their new dataset generation methods seem to essentially code for semantic relationships in a One-Hot coding way but instead of having each feature represented by a binary, it is represented by a list of binaries, which allows the authors to add noise to the representation of concepts? If I got that right, I think an explanation like that is going to make it easier to understand what the authors do and also what is new about their methods. This is also true when introducing their network. Authors discuss the whole idea of homeostasis in great detail but do not explain how their network is different from related (but well-known) network classes. In fact, the first time Hopefield networks are even mentioned is on page 6, even though they seem very relevant to the authors introduced architecture?

*Major: contextualisation of results*

The point above leads to an additional related issue that the authors should highlight in more detail how their findings actually go beyond the existing literature. I am not an expert in this class of networks, but their task seems to be very related to tasks already used since the early days of PDP (e.g. see McClelland and Rogers, Nat Revs Neuro, 2003) but they represented each feature through a list of binaries? At the same time for their findings in networks, I am under the impression that Hopfield networks and Boltzmann machines can also be configured to use either MLE or MAP (please correct me if I am wrong about this, as that might very well be the case)? As authors do not provide a description of what they think makes their model stand out against established alternatives, I find it difficult to judge how observing this in their model is special. My hunch would be that in Boltzmann machines one would have to bring in a prior through the weights but for the paper here it is induced by the updating, but authors should really clarify this themselves in the text and not leave it to the reader to figure this out.

*Major: presentation of results*

I would recommend the following changes to make it easier to follow / interpret results:
- Clearly state how many networks you train and what the dataset structure is
- Where variances are presented (like Fig4C), state whether these are standard errors / deviations or else
- Related to the deviation in 4C, in the text you seem to be interpreting the red line in 4C (Outgoing Dom) as being significantly different from the other lines, but the plotted variance suggest they might not be significantly different?
- Generally, results plots like 4D and 5E should report standard errors and significance tests
- Fig 5E seems to be a key analysis but the figure legend does not explain what the dots or bars actually refer to? I understand that authors want to argue that under different setup the different models mirror either the MAP or MLE prediction but to support that conclusion with a figure, they should present the expected pattern under each of these inference modes and then show how each of the models looks like either of these inference modes. As states above, the similarity should ideally be test for significance.

*Minor: stylistic changes / corrections*

- The in-text references should be in brackets, i.e. (O’Reilly, 2000) instead of O’Reilly (2000). The ICLR Latex file provides this as a citation option.
- Generally speaking, across all figures, I think authors could increase the font size relative to other visual elements and could then make the figures overall smaller to save space. Additional space could be used to explain the ideas better and contextualise findings, as discussed above.
- Figure 2B is already data / results recorded from the model, is that correct? In that case I find it confusing that it is labelled as ‘B’ and as such listed above the description of the actual network algorithm.
- Line 417: “are be asymmetric”, seems like there is an error there
- Line 303: “Now, we test whether these learnt weights can, during Test (recall) help recall during the test phase of the network.”, that sentence seems off?

---

> ### Author Response · Authors · 2024-11-28
>
> --> We are very thankful for the feedback received by reviewer nZZ9. We point reviewer nZZ9 to our overall comment (top) regarding general modifications made in the manuscript following the advice of all reviewers. Here is our point by point response to reviewer nZZ9:
>
>
> When introducing both their dataset and their model, I think it would help the general reader if they described their methodological advancements in context of established works that readers are likely familiar with. For example, their new dataset generation methods seem to essentially code for semantic relationships in a One-Hot coding way but instead of having each feature represented by a binary, it is represented by a list of binaries, which allows the authors to add noise to the representation of concepts? If I got that right, I think an explanation like that is going to make it easier to understand what the authors do and also what is new about their methods.
>
> --> We agree this is much more straightforward and clear, and have included this intuition in the main text:
>
>
>
> This is also true when introducing their network. Authors discuss the whole idea of homeostasis in great detail but do not explain how their network is different from related (but well-known) network classes. In fact, the first time Hopefield networks are even mentioned is on page 6, even though they seem very relevant to the authors introduced architecture?
>
> --> Thanks for pointing this out, this has also been included in Section 2.2 New
>
> The point above leads to an additional related issue that the authors should highlight in more detail how their findings actually go beyond the existing literature. I am not an expert in this class of networks, but their task seems to be very related to tasks already used since the early days of PDP (e.g. see McClelland and Rogers, Nat Revs Neuro, 2003) but they represented each feature through a list of binaries?
>
> --> Have highlighted this connection in the discussion
>
> At the same time for their findings in networks, I am under the impression that Hopfield networks and Boltzmann machines can also be configured to use either MLE or MAP (please correct me if I am wrong about this, as that might very well be the case)?
>
> --> We are not aware of any inductive bias or modification on the learning rules or architecture that would make a Hopfield Network or Boltzmann machine to perform MLE over MAP (or vice versa) when the correct outcome of each estimation is different. In our last section, we have given an example of input statistics where both estimations have a different outcome, and therefore tested the outcome for each of the different models.
>
> As authors do not provide a description of what they think makes their model stand out against established alternatives, I find it difficult to judge how observing this in their model is special. My hunch would be that in Boltzmann machines one would have to bring in a prior through the weights but for the paper here it is induced by the updating, but authors should really clarify this themselves in the text and not leave it to the reader to figure this out.
>
> --> Results have shown that, while a Boltzmann machine (only visible units) performs MAP as well as HBN (out), it has a poor performance at prototype learning (See New Fig. 4C).
>
> I would recommend the following changes to make it easier to follow / interpret results:
> Clearly state how many networks you train and what the dataset structure is
>
> --> We hope this is more clear in the new version
>
> Where variances are presented (like Fig4C), state whether these are standard errors / deviations or else
>
> -->Done in Fig. 4C and Table 2.
>
> Related to the deviation in 4C, in the text you seem to be interpreting the red line in 4C (Outgoing Dom) as being significantly different from the other lines, but the plotted variance suggest they might not be significantly different?
>
> --> In outgoing dom, the network is biased to recalling the overall most likely prototype, hence sometimes has a lower overlap with the original input (before noise) and a higher variance. However, precisely because it is more likely to appear overall, it does not affect average performance significantly.

---

> > ### Author Response · Authors · 2024-11-28
> >
> > Generally, results plots like 4D and 5E should report standard errors and significance tests
> > --> We did not have time to make significance test for Table 2. We would like to highlight that, rather than testing a hypothesis, we were quantifying the performance and bias towards MLE and MAP strategies, also obtaining an almost negligible trial-to-trial variability. However, if this is still an issue for the reviewer we would be happy to do so for the camera-ready version.
> >
> > Fig 5E seems to be a key analysis but the figure legend does not explain what the dots or bars actually refer to? I understand that authors want to argue that under different setup the different models mirror either the MAP or MLE prediction but to support that conclusion with a figure, they should present the expected pattern under each of these inference modes and then show how each of the models looks like either of these inference modes. As states above, the similarity should ideally be test for significance.
> > --> We hope the idea behind these metrics is better conveyed now.
> >
> >
> > Minor: stylistic changes / corrections
> > The in-text references should be in brackets, i.e. (O’Reilly, 2000) instead of O’Reilly (2000). The ICLR Latex file provides this as a citation option.
> > --> Done
> >
> > Generally speaking, across all figures, I think authors could increase the font size relative to other visual elements and could then make the figures overall smaller to save space. Additional space could be used to explain the ideas better and contextualise findings, as discussed above.
> > --> This is a very good point and has given us notably more space.
> >
> > Figure 2B is already data / results recorded from the model, is that correct? In that case I find it confusing that it is labelled as ‘B’ and as such listed above the description of the actual network algorithm.
> > --> Figure 2B can be considered “a schematic” rather than a result, as it just takes 3 different distributions (not real input fed to the network) to visually show the effect of the top-K operation.
> >
> > Line 417: “are be asymmetric”, seems like there is an error there
> > --> Thanks for spotting this, although this is no longer in the text.
> >
> > Line 303: “Now, we test whether these learnt weights can, during Test (recall) help recall during the test phase of the network.”, that sentence seems off?
> > --> Same

---

> > > ### Comment · Reviewer_nZZ9 · 2024-11-28
> > >
> > > Dear authors,
> > >
> > > thank you so much for engaging with my review of your manuscript. I think presenting results as they now are in Table 2 is helpful to understand the work but that brings me back to a point I made in my initial review that authors did not really address. Specifically, the main result of your work seems to be that you have this new network type that can be reconfigured to either perform like MLE or MAP and Table 2 shows how closely these two forms of predictions are matched. In my initial review I asked the question whether Hopfield networks / Boltzmann Machine could not do the same and from Table 2 it now seems like they actually can? At least the scores seem to be very close to the ones by the authors' model, for one of the models per prediction type? That is if a higher score means good in this table. I appreciate that the authors model can be reconfigured but that then seems to be the key contribution which should be stated very clearly in the key contributions?
> > >
> > > In my prior review I brought up the issue of contextualisation of their finding with the literature. They now mention at the start that they were inspired by Hopfield networks but given that a key result of their work is that their network can perform MAP / MLE, I think it should have been mentioned to which degree that was already possible before. Papers like this https://ieeexplore.ieee.org/abstract/document/374777 seem to study this in regular Hopfield networks. As said in my initial review, I am not an expert in this class of networks but there seems to be a rich history of studying these kind of networks in this environments and a clear manuscript would have presented prior results at the start, to then clearly outline how their current setup tries to go beyond that. This of course links to my prior paragraph.
> > >
> > > The authors say that they do not test for significances as they do not test a hypothesis. I do want to push back on this as when authors say that their model does something better than another model, then that is a specific hypothesis which should be tested. I know that for large scale neural networks that is not commonly done but the small architecture authors present here clearly allows for more rigour. Especially given that the results are very focused on Figure 4c and Table 2, so a relatively small set of analyses.
> > >
> > > Lastly, I find the choice of including a purely verbal link to testing neuroscience / cognitive sciences theories with their model an odd choice. They now mention this idea in their abstract and discussion but they do not actually use specific reference data, if I see this correctly.
> > >
> > > Overall, my prior review pointed out that the presentation of results are my main concern and I think that the manuscript does still not do a good job at clearly stating what the authors contribution are and how they link to what we already knew before. The findings as they are, I would still see as borderline regarding acceptance at the conference, but given that the presentation of results has not meaningfully improved, I would at this stage choose to keep my score as is.

---

> > > > ### Author Response · Authors · 2024-12-03
> > > >
> > > > Dear reviewer nZZ9,
> > > >
> > > > Thank you very much for your follow-up on the manuscript revision. We acknowledge many of the points raised by the reviewer, such as how the introduction/discussion could benefit from expanding in previous results regarding previous results MLE/MAP in models used in biology and the importance of statistical testing of hypotheses.
> > > >
> > > > However, we would like to also make the point that all the concerns raised seem to be regarding the last section of the manuscript (where connection to neuroscience and presentation of MAP/MLE results is made), the main goal of which we consider is not fully accounted for here. The motivation of section 2.4 is to showcase how our EGP could be used to test the validity of different models in neuroscience. In this sense, the schematic would be
> > > >
> > > > EGP --> task1 (prototype learnign) & task 2 (masked-input prediction) --> behavioural results --> compare/test against results shown in the manuscript).
> > > >
> > > > As this protocol is new, we don't have the data that would come from using it, so results here can be seen as a network-per-network prediction on two different tasks. However, we don't consider this is a verbal link, but instead a quantitative proposal of how to validate different models used in neuroscience using our protocol. This is the motivation there was behind not comparing networks between them, as these scores would instead need to be compared against data.
> > > >
> > > > The reviewer seems to emphasize that our main result is that HBNs can perform MAP/MLE, but we would like to highlight that are our main contributions, as stated in the manuscript, are:
> > > >
> > > > ***
> > > > • Proposing an Episode Generation Protocol (EGP) with an associated Semantic Structure,
> > > > which allows testing semantic learning in artificial neural networks.
> > > > • Presenting Homeostatic Binary Networks (HBNs). This simplification of biologically-
> > > > inspired neural networks allows a direct link between a formally defined Semantic Structure
> > > > and its learning dynamics.
> > > > • Using the EGP to obtain different behavioural signatures of a plethora of models used in
> > > > neuroscience, relating them to responses in prototype learning and masked input prediction
> > > > ***
> > > >
> > > > In this sense, we think one of our main contributions is our Episode Generation Protocol, which formalizes the notion of semantics of an input-sampling process. The protocol is not arbitrary but instead we show (see Appendix A.3) how it allows obtaining the semantic structure of previously used protocols that generate overlapping patterns in the literature. Similarly, we show how many different types of recurrent neural networks (different to HBNs) seem to align with our definition of semantic structure, obtaining recurrent weight matrices after learning with  correlations that go from 0.5 to 0.92 (Figure 2). This motivates the idea that our EGP and semantic structure are tightly connected to learning in neural networks.Then, we present HBNs as a model that pushes semantic learning up to virtually perfect alignment (>0.99) of weights with semantic structure. This means that it can store an internal model of input, which other networks struggle to. Finally, use use our EGP on many networks to obtain behavioral-signatures, that is, something to test the models against experiments. We have included the connection of HBNs to MAP/MLE as a minor results in the appendix, that we consider helps a mechanistic understanding of why HBNs perform so well in these tasks, but we would like to emphasize again this is not one of our main results.
> > > >
> > > > We have realized, however, that there is one hypotheses that we are implicitly testing, which is that HBNs are better at semantic learning (as measured by correlation of weights with semantic structure). Discussion with reviewer nZZ9 has helped motivate this and we would be happy to test for significance in the camera ready version.
> > > >
> > > > Overall, we would like to thank again the reviewer for engaging in the discussion and helping us improve our work!

---

### Official Review · Reviewer_F6ne · 2024-11-04

**Soundness:** 2
**Presentation:** 2
**Contribution:** 2
**Rating:** 5
**Confidence:** 3

**Summary:**

The paper proposes an Episode Generation Protocol (EGP) to generate vectors of binary numbers (each vector is an “episode”) such that every episode contains one concept per episode attribute. As a clarifying example, the authors consider the attributes to be place and food, and the concepts to be Italy, France, pizza, and croissant. The overlap between different binary episode-vectors is a key feature of these synthetically generated episodes and corresponds to shared concepts across episodes. The authors study how these episodes are learned by a model that includes adjustable inhibition, Hebbian learning, homeostatic plasticity, and short-term synaptic depression; and then study how these episodes are recalled from noisy inputs (pattern completion).

**Strengths:**

The authors propose a neural network model that touches on many areas of research across both biological and artificial systems. They write about its ability to perform pattern completion, Maximum A Priori Estimation (MAP), and Maximum Likelihood Estimation (MLE); while doing this with biologically inspired elements such as adjustable inhibition, Hebbian learning, homeostatic plasticity, and short-term synaptic depression. Given this breadth of scope, there are many connections that could be developed to either contribute to machine learning or to a better understanding of the brain.

**Weaknesses:**

The paper aims to make contributions across both biological and artificial systems but, while it is a good start, doesn’t fully connect to modern machine learning or to the brain through, for example, neural or behavioral data.

On the biological side, the authors propose a model that incorporates adjustable inhibition, Hebbian learning, homeostatic plasticity, and short-term synaptic depression. Given all of these ingredients are there predictions that can be made about behavior or neural activity? Or can the model recapitulate known experimental findings? For example, the authors referenced papers by Anna Schapiro so this may be a good contact point to compare the model and experimental findings.

On the machine learning side, the model as it is currently implemented seems more like a proof of principle. So I’m left wondering if this model can be applied to more realistic tasks or used to gain insight into modern machine learning models. Below are some suggestions.

* The authors compare the pattern completion abilities of their model to the Hopfield-Tsodyks model from 1988. How does the model connect to other more recent literature on pattern completion? For example, modern Hopfield networks. See, for example, Krotov 2023 “A new frontier for Hopfield networks” and Krotov & Hopfield 2016 “Dense associative memory for pattern recognition”.

* In the Introduction, the authors highlight an interesting distinction between 1) models where “each pattern separately represents a concept, and the overlap is a consequence of the concepts being semantically related”, and 2) models where “activity patterns correspond to the full content of episodes, and the overlaps between patterns represent common concepts in the episodes encoded by both patterns.”
My interpretation is that models in class 1 might correspond to neural networks trained on image classification datasets like MNIST or ImageNet where there is generally only a single central object in the image and the pixel values between different images encode some sense of similarity. Models in class 2 might correspond to neural networks trained on semantic segmentation where different images share common objects or “concepts” to use the terminology of the paper. Can the model in this paper be productively applied to datasets used to study semantic segmentation, and not just the toy dataset considered in the paper? Or, relatedly, can the ideas introduced in this paper be applied to networks that are used for semantic segmentation, for example, as a way of understanding the activity patterns in later layers of the network? At a more general level what I would like to see is some path forward for this model, to see the relevance on more realistic tasks.

* The authors make an interesting claim that their model performs Maximum A Priori Estimation (MAP) and Maximum Likelihood Estimation (MLE). It seems to me that demonstrating this would require more work in showing the neural network recapitulates the Bayesian computation.

**Questions:**

Please see the previous section for a number of questions and suggestions for improving the paper.

---

> ### Author Response · Authors · 2024-11-28
>
> --> We are very thankful for the feedback received by reviewer F6ne. We point reviewer F6ne to our overall comment (top) regarding general modifications made in the manuscript following the advice of all reviewers. Here is our point by point response to reviewer F6ne:
>
>
> The paper aims to make contributions across both biological and artificial systems but, while it is a good start, doesn’t fully connect to modern machine learning or to the brain through, for example, neural or behavioral data.
>
> --> We agree this is a very  valid point. Reviewer F6ne’s comments have prompted us to (i) rethink what novel contributions were already present in our initial manuscript but could be better conveyed and (ii) identify how our work could be better connected to neuroscience and ML. To achieve this,
>
> We start by summarizing the new results obtained:
>
> (1) to provide an example application of our EGP in ML, we have trained a feed-forward network using our protocol. While it is also a relatively simple architecture, our work connects with the work of McLelland (2003) and (2019), but highlights the ability of neural networks to extract semantics in a self-supervised setting.
>
> (2)  We have re-framed the last two sections as a section of "Behavioural Signatures of semantic learning" We hope this showcases how the EGP can be used to test competing models of learning in neuroscience.
>
> We  then have modified the results flow in this direction: 1. Inputs typically used to study semantic development in ML and computational neuroscience fail to formalize the causal relationships between semantic concepts, 2. We propose a novel framework that not only accounts for concept formation and has a clearly associated semantic structure 3. We test this input on classic models of recurrent neural networks, and obtain a novel result: while structures like MLP’s can uncover the semantic structure, the prominent biologically-constrained model based on hebbian learning, the Hopfield-Tsodkys connectivity, as well as a Boltzmann machine, fail to uncover these features. This is crucially predicted by the fact that both models rely on symmetric synapses 4. This motivates the proposal of an alternative model of learning that can capture asymmetric semantic structures: Homeostatic Binary Networks. 5. As a computational model of the nervous system, the model is tested against Hopfield-Tsodyks,  Boltzmann machine and Modern Hopfield models to perform prototype learning, yielding comparable performance to the top models 6. Next we study how different models behave under masked-input prediction paradigms. We also hope the idea of how the EGP connects to MAP and MLE is more clear, as well as the mathematical motivation of why HBNs excel at these.

---

> > ### Author Response · Authors · 2024-11-28
> >
> > On the biological side, the authors propose a model that incorporates adjustable inhibition, Hebbian learning, homeostatic plasticity, and short-term synaptic depression. Given all of these ingredients are there predictions that can be made about behavior or neural activity? Or can the model recapitulate known experimental findings? For example, the authors referenced papers by Anna Schapiro so this may be a good contact point to compare the model and experimental findings.
> >
> > -->We hope this is resolved now.
> >
> > On the machine learning side, the model as it is currently implemented seems more like a proof of principle. So I’m left wondering if this model can be applied to more realistic tasks or used to gain insight into modern machine learning models. Below are some suggestions.
> >
> > The authors compare the pattern completion abilities of their model to the Hopfield-Tsodyks model from 1988. How does the model connect to other more recent literature on pattern completion? For example, modern Hopfield networks. See, for example, Krotov 2023 “A new frontier for Hopfield networks” and Krotov & Hopfield 2016 “Dense associative memory for pattern recognition”.
> >
> > --> We have included results on Boltzmann Machines and Modern Hopfield Networks, and tested both on prototype learning and masked input prediction.
> >
> > In the Introduction, the authors highlight an interesting distinction between 1) models where “each pattern separately represents a concept, and the overlap is a consequence of the concepts being semantically related”, and 2) models where “activity patterns correspond to the full content of episodes, and the overlaps between patterns represent common concepts in the episodes encoded by both patterns.” My interpretation is that models in class 1 might correspond to neural networks trained on image classification datasets like MNIST or ImageNet where there is generally only a single central object in the image and the pixel values between different images encode some sense of similarity. Models in class 2 might correspond to neural networks trained on semantic segmentation where different images share common objects or “concepts” to use the terminology of the paper. Can the model in this paper be productively applied to datasets used to study semantic segmentation, and not just the toy dataset considered in the paper? Or, relatedly, can the ideas introduced in this paper be applied to networks that are used for semantic segmentation, for example, as a way of understanding the activity patterns in later layers of the network? At a more general level what I would like to see is some path forward for this model, to see the relevance on more realistic tasks.
> >
> > --> We have added a comment now how the EGP could be used in combination with standard benchmarks in the discussion
> >
> > The authors make an interesting claim that their model performs Maximum A Priori Estimation (MAP) and Maximum Likelihood Estimation (MLE). It seems to me that demonstrating this would require more work in showing the neural network recapitulates the Bayesian computation.
> >
> > --> We have included a section in the appendix where we show that, in the limit of small noise and outgoing homeostasis dominance, the network performs MAP over semantically-masked input. Similarly, in the limit of incoming homeostasis dominance the network performs MLE.

---

> ### Comment · Reviewer_F6ne · 2024-12-03
>
> Based on the comments and updates to the paper I have raised my score.

---

### Official Review · Reviewer_V4hF · 2024-11-04

**Soundness:** 3
**Presentation:** 2
**Contribution:** 2
**Rating:** 5
**Confidence:** 4

**Summary:**

The paper presents a method to encode the statistics of episodes in a memory network composed of binary units, connected with weights whose dynamics include Hebbian learning, homeostasis and synaptic depression. The first contribution is a method to  generate binary patterns encoding some semantic relationships defined as conditional probabilities on some finite space composed of concepts grouped in supersets called attributes (both constituting the episodes). The generation procedure consists of sampling the given concepts (one per attribute) and encoding them in a disjoint population of neurons. The network is then trained using some bespoke mechanisms for activity and synaptic homeostasis. The authors show how this process recovers the original semantic structure (defined as the conditional probabilities of concepts), performs pattern completion of concepts subject to noise; and claim that this process performs statistical estimation of missing or corrupted concepts in a given episode. They also present some mean field theory for the weights.

**Strengths:**

The paper shows an elegant reinterpretation of the semantic structure in episodes to be stored in memory. There are a variety of analyses that show the different, claimed results while keeping the explanatory complexity to the minimum. This makes this an attractive potential scientific theory that is interpretable from the beginning. The different mechanisms of the neural network are explained and length. The simplicity of the model is attractive but could be misleading in ways that I will explain later. I congratulate the authors for a great work, and the comments that follow are intended as my contribution to make this work stronger.

**Weaknesses:**

I will expose my concerns in no particular order:

- I am worried about the lack of explanation about what K is (I mean, the actual value). It seems to be the size of each of the groups that encode the concepts. In this sense, this might be very restrictive to make this model useful. I imagine that, if K < N, the pattern completion might not work as expected. It would be useful to show how the model behaves for different Ks
- I don't think the model contradicts Gastaldi et. al (2021). You still have a minimum amount of noise (which I interpret as the overlap), above which, the network can not recover. In this sense, I am not that sure the two schemas shown in figure 1A and 1B are that different, it seems that we have just renamed what the overlap is buy making those overlaps atomic units. This might be too strong of a constraint on the whole model, limiting the scalability.
- Regarding scalability, the simplicity of the exposition leaves some doubts about how much the model is able to scale (and therefore be of any scientific or engineering use), both in the complexity of the semantic structure (nested relations, higher order correlation?) and the complexity of the patterns (would that support different encodings).
- I have doubts about the strength of the conclusions drawn from the work in general, It is clear that the authors have a strong grasp of mathematics, however, I see no justification about the importance of each of the different mechanisms in achieving the ultimate result (homeostasis, depression). Even more, it seems that the Hebbian rule plus the top-K activation would unavoidably compute the covariance matrix of the input! (XX^T_ij = sum(d*xidxj))
- Connected to this, some of the mechanisms are not well justified, why is homeostasis and depression important? I suppose it is the renormalization of the weights but the theoretical justification (necessity and sufficiency) are week.
- The mean field analysis is interesting but also confusing. First, there is no coherent naming of the variables (sometimes Tout, Tpre), second it is not clear what the contribution to this analysis is to the full narrative of the paper (I can see the paper without the analysis and it would not change the end result). More on this later.
- There are a fair amount of typos in notation and in the text in general
- Figure 4 shows only 2 concepts for the attribute food but the caption says it is 4?
- Labels are wrong in figure 5D
- I disagree with the network performing MLE or MAP, or at least this claim should be qualified. Indeed, the model seems to oscillate! so, when do you know when to stop?
- Finally, a general comment. It would be nice to have a discussion about how plausible this model is based on what is known from the structure of memory.

In general, I am unsure about the relevance of the work in the context put forward by the authors.

**Questions:**

- One of the properties of episodic memory is the ordering of events and creation of chains. It is not immediately clear to me how this could be implemented (is, I think, what differentiate the problem from mere pattern completion/storage). I suggest including an experiment in this regard as your results (in order to justify calling it episodic, and to show that the model agrees with observations from the literature)

- The mean field theory seems to assume that the neurons are tracking the input statistics perfectly. I do not think you show that in the first place. The probability distribution you derive also seems to assume independence and disconnected neurons. Can you show this is a valid assumption, what are the implications of this assumptions for the rest of the claims of the paper.

- I did not see the phase transition in the weights that you mentioned explained in detail.

- The behaviour of the weights for the regimes with homeostasis does not seem to apply to the model (the dynamics are never reaching this regime). Can you explain your reasoning in more detail?



-

**Details Of Ethics Concerns:**

No concerns

---

> ### Author Response · Authors · 2024-11-13
> **Clarificiation regarding feedback**
>
> Dear Reviewer,
>
> Thank you very much for the thorough feedback. We are already working to improve the manuscript in line with your suggestions.
>
> In the meantime, we would like to ask for clarification on the following point:
>
> $\textbf{The behaviour of the weights for the regimes with homeostasis does not seem to apply to the model (the dynamics are never reaching this regime). Can you explain your reasoning in more detail?}$
>
> Could you please specify which figure or piece of results this comment refers to? Additionally, could you clarify which specific regime is not being reached and under what conditions? We have implemented three synaptic homeostatic regimes—outgoing homeostasis dominance, outgoing/incoming homeostatic balance, and incoming homeostasis dominance—each of which is imposed through the network parameters prior to training.
>
>
> Thank you very much in advance for your assistance!

---

> > ### Comment · Reviewer_V4hF · 2024-11-20
> > **The regimes of the weights achieved during the simulation**
> >
> > Thanks for your comment. Maybe I misunderstood the purpose of the mean field analysis but it seems that the homeostasis step clamps the weights at w_max^{in/out}. This means that the part of the analysis for which S_j^out >  w_max^{in/out} is never reached. I wondered then, if that is the case, about the relevance of those regimes for the dynamics of the network and the conclusions extracted from them about the connection between the joint probabilities and weights.

---

> > > ### Author Response · Authors · 2024-11-21
> > >
> > > Dear reviewer,
> > >
> > > Thank you very much for your response.
> > >
> > > Regarding the motivation behind the mean-field analysis (we will comment on this in more detail when we upload the official rebuttal), it is essentially two-fold (i) obtain the synaptic fixed points without need of simulations for other researchers interested in using the model and (ii) get a better intuition on how the weights reflect the input statistics and, ultimately, the associated semantic structure. We agree this was not highlighted enough and are working on improving the manuscript in this sense.
> > >
> > > Regarding this specific question, we will try to address it here but we are not sure we understood the concern, so please let us know if this helps clarify it.
> > >
> > > "the homeostasis step clamps the weights at w_max^{in/out}This means that the part of the analysis for which S_j^out > w_max^{in/out} is never reached"
> > >
> > > Indeed, synaptic homeostasis ensures that for, a given neuron, the total incoming weights (S^in) and outgoing weights (S^out), respectively, never surpass w_max^in and w_max^out. This is always true regardless of the model and stage of training.
> > >
> > > "I wondered then, if that is the case, about the relevance of those regimes for the dynamics of the network and the conclusions extracted from them about the connection between the joint probabilities and weights."
> > >
> > > Maybe we did not understand the question correctly, but in the mean-field analysis we always work under the conditions that S^in and S^out are at most the corresponding w_max values. Is there a particular equation or regime where your are not sure if the assumed conditions are necessarily true? Or is the question about what would happen in a model that did not have synaptic homeostasis?
> > >
> > > Thank you very much!

---

> > > > ### Author Response · Authors · 2024-11-28
> > > >
> > > > --> We are very thankful for the feedback received by reviewer V4hF. We point reviewer V4hF to our overall comment (top) regarding general modifications made in the manuscript following the advice of all reviewers. Here is our point by point response to reviewer V4hF:
> > > >
> > > > I am worried about the lack of explanation about what K is (I mean, the actual value). It seems to be the size of each of the groups that encode the concepts. In this sense, this might be very restrictive to make this model useful. I imagine that, if K < N, the pattern completion might not work as expected. It would be useful to show how the model behaves for different Ks
> > > > We thank the reviewer for pointing us at how the relation of K to the rest of parameters was not clear. We have made it more explicit in the text now, in sections 2.1 and 2.3.
> > > >
> > > > --> Indeed, we impose that $K$ represents both the number of active neurons per concept (in an episode, which we assume the reviewer is referring to as $N$) and the number of active neurons per region imposed by the activation function. Modifying K would be highly problematic as it would either decrease the signal (if decreased) or increased the noise (if increased). This is a limitation of the model, which in a more realistic would probably have to meta-learn the sparsity levels that maximize the signal to noise ratio. We have included a comment in the discussion.
> > > >
> > > > I don't think the model contradicts Gastaldi et. al (2021). You still have a minimum amount of noise (which I interpret as the overlap), above which, the network can not recover. In this sense, I am not that sure the two schemas shown in figure 1A and 1B are that different, it seems that we have just renamed what the overlap is buy making those overlaps atomic units. This might be too strong of a constraint on the whole model, limiting the scalability.
> > > >
> > > > --> We agree that the two views are not necessarily in conflict but are rather a reinterpretation of neural activity, concepts and episodes. The key difference is related to the comment made by the reviewer. You still have a minimum amount of noise (which I interpret as the overlap), above which, the network can not recover. We would like to highlight that, in our framework, consistent overlaps between episodes are themselves the concepts. In fact, in the absence of noise (Nswap = 0), we still can have highly overlapping inputs when they share a concept. For example, in the example shown in Fig.1, the input vector corresponding to an episode (Italy, pizza) and one corresponding to (Italy, croissant), would share 50% of active neurons. This would be opposed to episodes (Italy, pizza) and (France, croissant) that, due to noise, end up having shared neurons (this would also be the case in Gastaldi et al.l (2021)). The difference is that, in the case of Gastaldi et al. (2021), if two patterns have an overlap between over chance, that is interpreted as two different concepts being semantically related. Our interpretation is that for that to occur, there need to be at least 3 concepts, and that the overlapping fraction itself represents a concept in its totality.
> > > > To summarise, in Gastaldi et al. (2021)
> > > >
> > > > 1 pattern →  1 concept
> > > > If 2 patterns semantically unrelated → small overlap
> > > > If 2 patterns semantically related → high overlap (the overlap is much smaller than the concepts)
> > > > In our framework:
> > > > 1 pattern → 1 episode (a collection of concepts)
> > > > If 2 patterns semantically unrelated → small overlap
> > > > If 2 patterns semantically related (share a concept) → high overlap (the overlap is the shared concept)
> > > > Regarding scalability, the simplicity of the exposition leaves some doubts about how much the model is able to scale (and therefore be of any scientific or engineering use), both in the complexity of the semantic structure (nested relations, higher order correlation?) and the complexity of the patterns (would that support different encodings).
> > > > We would like to highlight that his model has its main use in modelling of neural systems. In this sense, the standard practice today are essentially hopfield networks. Additionally, these networks are usually tested on random quasi-orthogonal inputs, so we consider our input protocol is already a jump in complexity from modern training setups in computational modelling of biological neural networks. In this sense, we have shown that our model is the only one that can capture asymmetric semantic relationships between concepts.

---

> > > > > ### Author Response · Authors · 2024-11-28
> > > > >
> > > > > I have doubts about the strength of the conclusions drawn from the work in general, It is clear that the authors have a strong grasp of mathematics, however, I see no justification about the importance of each of the different mechanisms in achieving the ultimate result (homeostasis, depression).
> > > > >
> > > > > --> We have added an intuitive explanation in section 2.2 New
> > > > >
> > > > > Even more, it seems that the Hebbian rule plus the top-K activation would unavoidably compute the covariance matrix of the input! (XX^T_ij = sum(d*xidxj))
> > > > >
> > > > > -->Yes, that would be the case in the absence of homeostatic plasticity / synaptic competition (our network always eventually starts implementing it at some point)
> > > > >
> > > > > Connected to this, some of the mechanisms are not well justified, why is homeostasis and depression important? I suppose it is the renormalization of the weights but the theoretical justification (necessity and sufficiency) are week.
> > > > > The mean field analysis is interesting but also confusing. First, there is no coherent naming of the variables (sometimes Tout, Tpre), second it is not clear what the contribution to this analysis is to the full narrative of the paper (I can see the paper without the analysis and it would not change the end result). More on this later.
> > > > >
> > > > > -->We would like to reply to these two points together.
> > > > > As the reviewer suggests, normalization plays a key role in transforming the hebbian updates, which reflect joint firing probabilities, into conditional firing probabilities, due to the multiplicative renormalization that relates to the marginal firing probability of the pre- or post- neuron depending on the homesostatic regime. Regarding sufficiency, we hope that is clear from the analytical results. Regarding necessity, it could very well be that other forms of learning induced similar synaptic fixed points, but that would mean that the mechanism fundamentally approximates the one highlighted here.
> > > > > Given that the model both captures biological properties and shows unique computational capabilities, we consider that publishing the analytical trajectories followed can be of general interest for researchers using the model. By using the derived fixed points, one can predict the network structure without actually having to train it. Similarly, we hoped for the derivation to help get an intuition of the role played by the different network components.
> > > > > At the same time, we agree that the justification was not clear, and we have included an intuition and connection to the theoretical results in New Section 2.2
> > > > >
> > > > > There are a fair amount of typos in notation and in the text in general
> > > > > --> We apologize for that and will do a thorough proofreading before the camera-ready version.
> > > > >
> > > > > Figure 4 shows only 2 concepts for the attribute food but the caption says it is 4?
> > > > > --> Thank you very much for spotting this. However, this figure is no longer in the new version.
> > > > >
> > > > > Labels are wrong in figure 5D
> > > > > --> Also thank you very much for pointing this out, although this figure has also been removed.
> > > > >
> > > > > I disagree with the network performing MLE or MAP, or at least this claim should be qualified. Indeed, the model seems to oscillate! so, when do you know when to stop?
> > > > > -->Regarding the oscillatory behaviour of the model, we are not sure about the concern. In the training phase, the model reaches stable weights (Fig. 3C). If the concern is about why the performance under MAP or MLE paradigms is distributed instead of being constant, this is due to the noise injected into the pattern at each trial.
> > > > > We have made substantial changes to this figure, which we hope now is better explained. Similarly, we have added a section in the appendix where show the theoretical connection between the learned weights and MLE/MAP
> > > > >
> > > > > Finally, a general comment. It would be nice to have a discussion about how plausible this model is based on what is known from the structure of memory.
> > > > > -->We have added a comment on this in the discussion.
> > > > >
> > > > > In general, I am unsure about the relevance of the work in the context put forward by the authors.
> > > > > We hope the new manuscript better conveys the relevance of our work. We summarize what we consider are its main contributions:
> > > > > 1-A formalizing a protocol for input generation to study semantic learning. We now have shown how this can be used to systematically study many different types of RNNs, exploring to what degree different networks extract semantics from input patterns.
> > > > > 2- Proposing a model that outperforms in semantic learning classic neural networks used to model neural systems. Crucially, it also shows computational advantages when performing predictions about uncertain environments under different predictive regimes.
> > > > > 3- Combined, shown how our EGP can be used to compare the behavioural output of biological agents to neural networks, providing a way to make models compete as mechanisms for semantic learning in neural systems.

---

> > > > > > ### Author Response · Authors · 2024-11-28
> > > > > >
> > > > > > Questions:
> > > > > > One of the properties of episodic memory is the ordering of events and creation of chains. It is not immediately clear to me how this could be implemented (is, I think, what differentiate the problem from mere pattern completion/storage). I suggest including an experiment in this regard as your results (in order to justify calling it episodic, and to show that the model agrees with observations from the literature)
> > > > > > --> We acknowledge this is a model of episodic memory where episodes have already been compressed in time. In this sense, there is no temporal correlation between episodes. We have included a comment on this in the discussion.
> > > > > >
> > > > > >
> > > > > > The mean field theory seems to assume that the neurons are tracking the input statistics perfectly. I do not think you show that in the first place. The probability distribution you derive also seems to assume independence and disconnected neurons. Can you show this is a valid assumption, what are the implications of this assumptions for the rest of the claims of the paper.
> > > > > > --> In the mean-field analysis, we do consider neurons to perfectly track input statistics. We consider this assumption is realistic during the learning regime, where the signal to noise ratio is small and the network is allowed to “believe” its input (this is a standard practice in computational modelling of neuroscience and in machine learning, many times noisy-less input is fed to the network). However, it should be noted that we are already adding noise in the input through the episode input mapping, which randomly swaps neuronal activity. Our framework would be identical to having clean input where each input neuron unequivocally reflects the concepts present in the episode, and then making the activity of network neurons noisy. However, this allows us to decouple the mean field analysis, where neurons are allowed to perfectly track input statistics, and the analysis of the input distribution, which of course then gets more complicated (derived in Appendix C). It should be additionally noted that, in that analysis, there is not independence between neurons, as activity is restricted by the top-K operation and not all configurations of neurons are possible. We would like to highlight, too, the almost exact matching of simulations and theory in the limits where the fixed points are derived (dominating outgoing or incoming homeostasis), as reflected in Fig. 3C. In particular, the differences between same-coloured curves are well accounted for, with these differences being due to the Nswap operations.
> > > > > > We also make the assumption of absence of physical connections (disconnected) neurons during learning, which is again motivated by learning periods in which the internal model (recurrent synapses) are ignored and the neurons are input-driven instead. This is also a standard practice in most biologically-motivated networks, and is in fact something used in standard hopfield networks that separate pattern storage and retrieval.
> > > > > > We have however added a comment on this in the discussion.
> > > > > >
> > > > > > I did not see the phase transition in the weights that you mentioned explained in detail.
> > > > > > --> We have added a clarification in the main text.
> > > > > >
> > > > > > The behaviour of the weights for the regimes with homeostasis does not seem to apply to the model (the dynamics are never reaching this regime). Can you explain your reasoning in more detail?
> > > > > > --> We are still not sure about this question (please see our last clarification comment).

---

### Official Review · Reviewer_6ryh · 2024-11-07

**Soundness:** 3
**Presentation:** 1
**Contribution:** 3
**Rating:** 6
**Confidence:** 3

**Summary:**

The literature suggests the overlap in neural activity as a separation of meaningful semantic concepts, where the overlap is defined to be between episodic patterns. The literature proposed here considers activity patterns to correspond to the full content of episodes, and the overlaps between patterns represent common concepts in the episodes encoded by both patterns. Thus, the Episode Generation Protocol (EGP) system is introduced to map the semantic structures of episodes to input pattern generation. By using the inputs produced by the EGP, the literature uses a Homeostatic Binary Network as a method to train and recall episodes. These are motivated by the fact that the trained network will be capable of recalling even if the episode has concepts that are overlapping towards other attributes due to the semantic structure provided by the EGP.

**Strengths:**

The idea of using a semantic structure to define episodes based on a structure of concepts and attributes is great. What I really like about the paper is how this structure is used to train a network for recalling patterns, which points its focus towards the structure of the input rather than training the network to learn. This key idea is novel.

**Weaknesses:**

I do understand that some of the technical details to appendix helps shorten and simplify the explanation in the main text. I believe some of them are way too shortened on formal explanation and way too lengthened on an intuitive explanation, e.g. key concepts such as explanation of EGP is good but it would benefit from having explanations such as how the episode is mapped to an input would help. Another thing to note here is that instead of giving an intuitive explanation, it would be better to connect a formal explanation to some extent that refers to both the appendix and an intuitive explanation would provide a better understanding of the topic.
Section structured for 2 is confusing. I find the methodology and results to be mixed. The issue with the section is it is difficult to understand the motivation of the paper and its concept if I have not read the paper fully. I believe the paper would benefit if these are laid out in the methodology with the results (probably 2.4 and 2.5?) in a separate section. The issue mainly persists in the methodology part of it.

**Questions:**

Figure 1C (middle): I believe that the probability for the attribute relation between italy and pizza may have an error? (should be 0.25?)
I do understand that to add some extra variability, $N_swap/2$ neurons are swapped active neurons with the inactive neurons. What does $N_swap$ in this case?
A.1: During explanation of eq (11),  does ‘a’ here represent an episode concept? Since an episode contains one episode concept per episode attribute, A is an episode attribute and thus ‘a’ is one episode concept rather than ‘a’ being a set of episode concepts. This may need clarification.
A.2 The definition of i is a population of associated neurons as SEN_a as per episode-input mapping. Here, you define it as an episode concept? I believe the terms here need to be further clarified since it is very confusing.
A.3.1: The definition of satellite objects and its relation to its features is key here. This needs to be cleared out to better explain the readers on EGP.
Consider explaining the input mapping at the start. In my opinion it seems unclear until after going through the appendix.
Section 2.2: The definition of top-K activation should define what region specifically means (neurons associated to attribute?)
Although the experiment shown here considers a short episode, I would also evaluate its performance on longer episodes (with more swaps) and consider how this system may perform.

---

> ### Author Response · Authors · 2024-11-13
> **Clarificiation regarding feedback**
>
> Dear Reviewer,
>
> Thank you for your detailed feedback. We are now actively implementing your suggestions to enhance the manuscript.
>
> In the meantime, we would like to ask for a few clarifications:
>
> ****1****
>       "Section structured for 2 is confusing. I find the methodology and results to be mixed. The issue with the section is it is difficult to understand the motivation of the paper and its concept if I have not read the paper fully. I believe the paper would benefit if these are laid out in the methodology with the results (probably 2.4 and 2.5?) in a separate section.”
>
> Just to confirm, are you suggesting to have a "Section 2 - Methods" (including current 2.1 and 2.2) and then leave for "Section 3 - Results" the current 2.3, 2.4, and 2.5? We do think that would be a great idea and acknowledge that the current format might be confusing.
>
> ***2***
>      "Figure 1C (middle): I believe that the probability for the attribute relation between italy and pizza may have an error? (should be 0.25?)"
>
> We believe the probability shown is correct, but we would like to double-check with you in case we missed something! In Figure 1C (middle), we set the probability of the episode (Italy, pizza) to be 0.25. (shown in Figure). This probability is imposed for illustration and could have been assigned any other value. If you are referring to Fig. 1D, the conditional probability of pizza given Italy is 1, as all episodes containing Italy also contain pizza. The conditional probability of Italy given pizza is 0.5, as the probability of (France, pizza) is also 0.25, so half of the episodes that contain pizza contain Italy, and the other half France. Does this seem correct?
>
> ***3***
>      “I do understand that to add some extra variability,neurons are swapped active neurons with the inactive neurons. What does in this case?”
>
> Could you please clarify this question? Indeed, after each neuron coding for an element is activated if that element is contained in the episode. Then, noise is injected by silencing a fraction of active neurons, and the same number of inactive neurons are activated. Is the question regarding the motivation for doing this?
>
> ***4***
>      “Although the experiment shown here considers a short episode, I would also evaluate    its performance on longer episodes (with more swaps) and consider how this system may perform.”
>
> To confirm, are you suggesting that during training we present the same input pattern across multiple steps, each with a different set of neurons swapped? If so, we would expect similar behaviour, as the number of training steps (5000) is very large compared to the combinations of episode elements, meaning the network already encounters many independent samples of each episode. However, we could run this experiment and add a supplementary figure.

---

> ### Comment · Reviewer_6ryh · 2024-11-21
>
> Thank you for your comments.
> Section 2.1, 2.2 and 2.3 can be set as methodology with some substantial information from the appendix in it would help (For e.g. A.2 where the motivation behind your architecture and explanations on Bussmann et al. (2006)).
> Its on Figure 1C. I suspect that fig 1 C. P(Italy,Croissant) is wrong since it shows 0. in the figures. Please correct me if I am wrong.
> Yes its on the motivation behind why you used (N_swap/2). Is N_swap considered a parameter where 0 < N_swap <= Total number of neurons for sensory layer where N_swap is divided by 2 so there can always be the same number of swaps from 0 to 1 and vice versa?
> As for the last one i was assuming that increasing N_swap (specifically on larger ones) for noisier signals and consider how different timesteps of training may affect its recall performance (possibly multiple repetitions of the experiment instead?). This is to further report whether the better performance in higher noise may be attributed to either overfitting to noise/signals or just recognizing the pattern itself.

---

> > ### Author Response · Authors · 2024-11-28
> >
> > --> We are very thankful for the feedback received by reviewer 6ryh. We point reviewer 6ryh to our overall comment (top) regarding general modifications made in the manuscript following the advice of all reviewers. Here is our point by point response to reviewer 6ryh:
> >
> > I do understand that some of the technical details to appendix helps shorten and simplify the explanation in the main text. I believe some of them are way too shortened on formal explanation and way too lengthened on an intuitive explanation, e.g. key concepts such as explanation of EGP is good but it would benefit from having explanations such as how the episode is mapped to an input would help.
> >
> > --> Following this suggestion and that of Reviewer nZZ9, we have expanded on the nature of the episode-input mapping.
> >
> > Another thing to note here is that instead of giving an intuitive explanation, it would be better to connect a formal explanation to some extent that refers to both the appendix and an intuitive explanation would provide a better understanding of the topic.
> >
> > --> Guided by this comment, and that of reviewer V4hF, we have included the connection of homeostatic plasticity → weight trajectories → semantic structure in the main text (section 2.2 New).
> >
> > Section structured for 2 is confusing. I find the methodology and results to be mixed. The issue with the section is it is difficult to understand the motivation of the paper and its concept if I have not read the paper fully. I believe the paper would benefit if these are laid out in the methodology with the results (probably 2.4 and 2.5?) in a separate section. The issue mainly persists in the methodology part of it.
> >
> > --> We hope the flow of the paper is smoother in the new version, which goes like (EGP & Semantic Structure → Test impact of Semantic Structure in Classic RNNs → Propose HBN as a model that fully extracts semantics → Behavioural comparison of all models).
> >
> > Questions:
> > Figure 1C (middle): I believe that the probability for the attribute relation between italy and pizza may have an error? (should be 0.25?)
> >
> > --> After live discussion, we would like to highlight that probabilities in Fig. 1C “cannot be wrong”, as those are simply the imposed input statistics! In this case we are giving an example where the joint probability of Italy and croissant is 0, which is totally fine as long as all episode probabilities add up to one. What could be wrong would be the conditional likelihoods derived in Fig. 1D, but we have checked them and they are consistent with the probabilities in Fig. 1C. We hope this explains the rationale of these values but if something is still unclear please let us know.
> >
> > I do understand that to add some extra variability,neurons are swapped active neurons with the inactive neurons. What does in this case?
> >
> > --> We have refined the explanation of the Nswap operation in section 2.1 and Appendix A.
> >
> >  A.1: During explanation of eq (11), does ‘a’ here represent an episode concept? Since an episode contains one episode concept per episode attribute, A is an episode attribute and thus ‘a’ is one episode concept rather than ‘a’ being a set of episode concepts. This may need clarification.
> >
> > --> To answer your question, “a” is indeed an episode concept, and it is included in the episode attribute A (which is a set).  Was there a confusion regarding the phrase “Each episode attribute representing a set of episode concepts a ∈ A, b ∈ B, c ∈ C”? The idea here was that each attribute (A, B, and C) represents a separate set of concepts that are named with lowercase (a ∈ A, b ∈ B, c ∈ C). However, to address any possible confusions in this section, we have rephrased it and included how eq. 11 looks like in our particular case of place and food to give a more concrete example.
> >
> > A.2 The definition of i is a population of associated neurons as SEN_a as per episode-input mapping. Here, you define it as an episode concept? I believe the terms here need to be further clarified since it is very confusing.
> >
> > --> We have rephrased this section and again used the intuitive proposal of reviewer nZZ9 of “a list of One-Hot encodings”, which allows us to get rid of the maths.
> >
> >  A.3.1: The definition of satellite objects and its relation to its features is key here. This needs to be cleared out to better explain the readers on EGP. Consider explaining the input mapping at the start. In my opinion it seems unclear until after going through the appendix.
> >
> > --> We have rephrased this section and emphasized the episode-input mapping. We hope this is more clear now.

---

> > > ### Author Response · Authors · 2024-11-28
> > >
> > > Section 2.2: The definition of top-K activation should define what region specifically means (neurons associated to attribute?)
> > >
> > > --> We have made this explicit in (Section 2.2 New) and Eqs. 6, 10.
> > >
> > > Although the experiment shown here considers a short episode, I would also evaluate its performance on longer episodes (with more swaps) and consider how this system may perform.
> > > *As for the last one i was assuming that increasing N_swap (specifically on larger ones) for noisier signals and consider how different timesteps of training may affect its recall performance (possibly multiple repetitions of the experiment instead?). This is to further report whether the better performance in higher noise may be attributed to either overfitting to noise/signals or just recognizing the pattern itself.
> > >
> > > --> If we understand correctly, this is (and was) shown in Fig. 4C (new), but please let us know if that’s not the case and we can still add another supplementary figure in the camera ready version, since this was asked in the initial review.

---

### Author Response · Authors · 2024-11-28
**Summary of changes done in manuscript after revision**

We are very thankful to all reviewers for their thorough feedback, which has led to substantial modifications in the original manuscript. We would like to indicate here two major concerns raised by reviewers and how these have been addressed in the revised version.

Presentation and motivation for the work presented:
We have done an effort to improve the figures, readability and overall flow of the story. Sections have been re-structured and now read as follows: 1- Presentation of EGP, 2- Study of Semantic Learning in classic models of RNNs, 3- Proposal of HBNs as a model that can align better than tested models its internal model to the semantic structure, 4-Use of our EGP as a method to obtain behavioural signatures of HBNs and other models used in neuroscience.

Weak connection to modern AI or neuroscience:
We hope to give evidence of how our EGP can be used to systematically test semantic learning in artificial neural networks. To do this, we have extended our experiments to various models such as Boltzmann machines, Modern Hopfield Networks and feed-forward networks. Similarly, we showcase how the protocol can be used to study semantic learning in humans or animals, as well as how to use experimental behavioural responses to test different models used in neuroscience. HBNs themselves have different computational capabilities (comparable performance at prototype learning, better performance at semantic extraction, better performance at MAP/MLE prediction) that make them useful as models in neuroscience.

---

### Meta-Review · Area_Chair_N8ax · 2024-12-23

**Metareview:**

This paper explores a perspective on the potential role of overlapping representations for the encoding of semantic relationships between concepts. To do so the authors propose an Episode Generation Protocol (EGP) to generate vectors of binary numbers (each vector is an “episode”) such that every episode contains one concept per episode attribute. The authors study how these episodes are learned by a model that includes adjustable inhibition, Hebbian learning, homeostatic plasticity, and short-term synaptic depression; and then study how these episodes are recalled from noisy inputs (pattern completion).

Strengths: The paper proposes an elegant idea of using a semantic structure to define episodes based on a structure of concepts and attributes. The proposed neural network model touches on many areas of research across both biological and artificial systems and can perform pattern completion, Maximum A Priori Estimation (MAP), and Maximum Likelihood Estimation (MLE); while doing this with biologically inspired elements such as adjustable inhibition, Hebbian learning, homeostatic plasticity, and short-term synaptic depression.

Weaknesses: The main weakness of this paper is its clarity of presentation. Each reviewer wrote quite a different summary of the paper, and each reviewer raised concerns and questions about the clarity. While some of these points have been addressed during the discussion it is my impression that this issue remains unresolved.

Overall I recommend rejecting this paper, but I would encourage the authors to further distill the core of their contribution, improve the presentation and resubmit.

**Additional Comments On Reviewer Discussion:**

There was some back and forth during the discussion mostly regarding the clarity and structure of the paper. My impression is that the presentation was improved, but is still not up to the standards of an ICLR paper.

---

### Decision · Program_Chairs · 2025-01-22

Reject